# Integrated multi-omics reveals cellular and molecular interactions governing the invasive niche of basal cell carcinoma

Laura Yerly [1], Christine Pich-Bavastro [1], Jeremy Di Domizio [1], Tania Wyss[2], Stéphanie Tissot-Renaud[3], Michael Cangkrama[4], Michel Gilliet [1], Sabine Werner [4] & François Kuonen [1] ✉

Tumors invade the surrounding tissues to progress, but the heterogeneity of cell types at the tumor-stroma interface and the complexity of their potential interactions hampered mechanistic insight required for efficient therapeutic targeting. Here, combining single-cell and spatial transcriptomics on human basal cell carcinomas, we define the cellular contributors of tumor progression. In the invasive niche, tumor cells exhibit a collective migration phenotype, characterized by the expression of cell-cell junction complexes. In physical proximity, we identify cancer-associated fibroblasts with extracellular matrix-remodeling features. Tumor cells strongly express the cytokine Activin A, and increased Activin A-induced gene signature is found in adjacent cancer-associated fibroblast subpopulations. Altogether, our data identify the cell populations and their transcriptional reprogramming contributing to the spatial organization of the basal cell carcinoma invasive niche. They also demonstrate the power of integrated spatial and single-cell multi-omics to decipher cancer-specific invasive properties and develop targeted therapies.

Tumor cell invasion into the surrounding tissue is a hallmark of cancer[1,2]. By hijacking cellular and molecular processes governing normal tissue morphogenesis, tumor cells migrate into tissues to escape starvation. Moving towards a favorable environment ultimately leads to dissemination throughout the body, a process called metastasis[1]. As a part of tumor survival and progression, various mechanisms of tumor migration, dependent on specific cell–cell interactions and/or cell–substrate adhesion, have been reported[3]. Mesenchymal tumor migration typically relies on strong cell adhesion to the extracellular matrix (ECM) via focal adhesion complexes and associated stress fibers, leading to transient protrusion[4]. Ameboid tumor migration is rather driven by actomyosin or bleb protrusions with low ECM adhesion, allowing fast and versatile migration[4,5]. By contrast, collective migration of epithelial tumors relies on strong cell–

cell adhesions, allowing migration as clusters, sheets, or strands within surrounding tissues[6]. Importantly, these hierarchical cell motility modes can be finely tuned by cell–matrix (single-cell migration mode) or cell–cell (collective migration mode) adhesion forces[7]. In fact, as a result of the reciprocal reprogramming of tumor cells and stromal components[8], motility may adapt. The resulting plasticity favors tumor progression and resistance to treatment, emphasizing the need to further understand the complex biology of tumor–stroma interactions[8–10]. However, given the heterogeneity of cell types involved at the leading edge, previous studies using traditional experimental techniques have lacked spatial information or suffered from insufficient cell resolution.

Basal cell carcinomas (BCCs), which are the most frequent type of cancer in humans[11], offer an attractive model to address the role of

[1]Department of Dermatology and Venereology, Hôpital de Beaumont, Lausanne University Hospital Center, 1011 Lausanne, Switzerland. [2]Bioinformatics Core Facility (BCF), Swiss Institute of Bioinformatics, Quartier UniL-Sorge, Bâtiment Amphipole, 1015 Lausanne, Switzerland. [3]Department of Oncology, Immune Landscape Laboratory, Center of Experimental Therapeutics, Lausanne University Hospital Center, 1011 Lausanne, Switzerland. [4]Institute of Molecular Health Sciences, Department of Biology, ETH Zurich, 8093 Zürich, Switzerland. ✉e-mail: francois.kuonen@chuv.ch

reciprocal tumor-stroma interactions in tumor invasion, since they display both a large clinical and histopathological spectrum, despite genetic homogeneity[12,13]. Superficial and nodular BCCs are considered low-risk tumors, whereas infiltrative BCCs have a higher risk of progression[14]. The latter is associated with a higher recurrence rate after surgery[15], lower response to radiotherapy[16,17], and rare but fatal metastatic dissemination[18,19]. Classically, nodular and infiltrative BCCs are primarily distinguished by the histological morphology of their respective tumor-stroma interface. Nodular morphology is characterized by tumor nests that are demarcated from a loose myxoid stroma by a well-defined basement membrane and peritumoral cleft suggesting poor tumor matrix adhesion[11,20]. By contrast, infiltrative morphology is characterized by irregular tumor strands within a densely packed fibrotic stroma (hence the "sclerodermiform" denomination), where the basement membrane and peritumoral cleft are mostly absent[11,20]. Importantly, nodular and infiltrative morphologies found at the tumor-stroma interface are not categorical entities but rather represent a continuum reflected by both the inter- and intra-tumor heterogeneity. Therefore, BCCs offer the intriguing possibility to study the dynamics of transcriptional reprogramming at the tumor-stroma interface required for cancer invasion.

Here, we integrate single-cell RNA sequencing (scRNA-seq) transcriptomes from infiltrative BCCs with spatially-resolved transcriptomes obtained from BCC tumor and adjacent stroma areas. Thereby, we identify the spatial transcriptional reprogramming at the leading edge niche, where tumor cells with collective migration feature co-localize with ECM-remodeling fibroblasts. Importantly, we identify Activin A as a paracrine-acting factor regulating the transcriptional crosstalk and spatial organization of the invasive niche.

## Results

### scRNA-seq resolves the heterogeneous cell composition of infiltrative BCCs

To gain insight into the cellular and molecular mechanisms that govern tumor invasion in skin cancer, we first performed scRNA-seq on fresh biopsies of infiltrative BCCs collected from five individuals (Supplementary Fig. 1a, b). Tumor tissues were enzymatically digested into single-cell suspension followed by FACS sorting of living single cells, and scRNA-seq using the 10× Genomic platform. After quality control and removal of low-quality cells and doublets, a total of 28,810 cells were retained for further analyses (Fig. 1a and Supplementary Fig 1c). Once the cells were clustered, they were annotated to major cell types according to the expression level of canonical marker genes[21–23] (Fig. 1b, c). Overall, ten major cell types were characterized: epithelial cells, T cells, melanocytes, myeloid cells, endothelial cells, fibroblasts, cycling cells, B cells, pericytes, and mast cells (Fig. 1b, c). Epithelial cells represented the majority of cells (50.63%, Supplementary Fig. 1d) followed by immune T cells (22.79%, Supplementary Fig. 1d). Fibroblasts (2.81%), pericytes (1.69%) and endothelial cells (3.27%) were less abundant, potentially due to the suboptimal single-cell dissociation[24,25] (Supplementary Fig. 1d). Except for B cells, the 10 distinct cell types were represented in all 5 samples (Fig. 1d, e and Supplementary Fig. 1e, f) and were consistent with data obtained from scRNA-seq studies of other solid tumors[21,26,27]. To identify tumor cells, epithelial and T cells were subclustered (Fig. 1f) and analyzed for the chromosomal landscape of inferred copy number variations (using InferCNV[28]). When using T cells as a reference, the epithelial cluster C0 displayed higher copy number variations compared to the epithelial clusters C1-C2-C3 in all five samples (Fig. 1g, Supplementary Fig. 2). In particular, when excluding gene expression changes commonly observed in all five BCC samples (and thus potentially reflecting cell lineage-dependent transcription rather than genomic structural changes), inferred CNVs were exclusively found in the epithelial cluster C0, with a few being shared by virtually all cells of the cluster.

Consistently, BCC markers (*PTCH1*, *GLI1*, *GLI2*, *HHIP*, and *MYCN*) as well as Hedgehog signaling pathway and BCC KEGG signatures were enriched in cluster C0 when compared to clusters C1, C2 and C3 (and T cells) (Fig. 1h and Supplementary Fig. 3a–b). In contrast, cluster C3 showed specific enrichment for hair follicle signatures[29] (Supplementary Fig. 3c), cluster C1 gradual enrichment for basal epidermis signatures[29] (Supplementary Fig. 3d), and cluster C2 gradual enrichment for spinous/granular epidermis signatures[29] (Supplementary Fig. 3e), consistently with their normal keratinocyte nature. Based on combined Seurat clustering and inferred CNV analysis, we thus identified epithelial cluster C0 as tumor cells, while epithelial clusters C1, C2, and C3 represent basal, differentiated, and hair follicle keratinocytes, respectively (Fig. 1f). Taken together, these data provide a representative picture of infiltrative BCC composition.

### Spatial transcriptomics defines compartment-specific signatures for nodular and infiltrative tumor–stroma interfaces

scRNA-seq highlights the cellular and molecular heterogeneity of the tumor tissue as a whole but lacks information about the relevant tumor–stroma interactions at the invasive front of the tumor. To address this issue, we used digital spatial profiling (DSP) technology from GeoMx[30] (Nanostring) to selectively target 24 regions of interest (ROIs) covering tumor-stroma interfaces with infiltrative morphology (distributed across 6 infiltrative BCC samples) (Supplementary Fig. 1a). Twenty-four ROIs covering tumor–stroma interfaces with nodular morphology (distributed across 6 nodular BCC samples) were used as controls (Supplementary Fig. 1a). Each ROI was then divided into tumor area of interests (AOIs) based on pan-cytokeratin positivity (panCK$^{pos}$) and stroma AOIs based on panCK negativity (panCK$^{neg}$) (Fig. 2a and Supplementary Fig. 4a), and individually sequenced using the cancer transcriptome atlas library (CTA, 1812 genes). Importantly, after quality control and normalization, the number of counts per AOI was not affected by the tumor type or the scanning process (Supplementary Fig. 4b). To establish specific signatures reflecting tumor-stroma morphology, we then compared the differentially expressed genes (DEGs) in tumor and stroma AOIs between infiltrative and nodular BCCs. A total of 86 genes were differentially expressed between the infiltrative and nodular tumor AOIs (Fig. 2b, Supplementary Data 1, and Supplementary Fig. 4c), and 52 genes were differentially expressed between the infiltrative and nodular stroma AOIs (Fig. 2c, Supplementary Data 2, and Supplementary Fig. 4c). Consistently, the expression profile of the genes included in our DEG signature (138 genes) segregated tumor nodular (T$^{NOD}$), tumor infiltrative (T$^{INF}$), stroma nodular (S$^{NOD}$) and stroma infiltrative (S$^{INF}$) AOIs (Supplementary Fig. 4d). Intriguingly, T$^{INF}$ AOIs appeared closer to stroma AOIs compared to T$^{NOD}$ AOIs, while S$^{NOD}$ AOIs appeared closer to tumor AOIs compared to S$^{INF}$ AOIs, suggesting that T$^{INF}$ and S$^{INF}$, as well as T$^{NOD}$ and S$^{NOD}$ transcriptomes respectively, may cross-contaminate each other. Indeed, when overlapping T$^{NOD}$ and S$^{NOD}$ DEGs as well as T$^{INF}$ and S$^{INF}$ DEGs, we observed 5 and 18 shared genes, respectively (Supplementary Fig. 4e). To avoid that cross-contamination affects spatial signatures, we thus filtered the DEGs using scRNA-seq data from *KRT14*$^{pos}$ clusters (for tumor AOIs) or *KRT14*$^{neg}$ clusters (for stroma AOIs). *FN1*, for example, was not expressed in *KRT14*$^{pos}$ clusters based on scRNA-seq (Supplementary Fig. 4f, upper panels). Consistently, fluorescence in situ hybridization (FISH) using RNAscope® analysis for *FN1* on infiltrative BCC confirmed the exclusive expression of this gene in the stromal compartment (Supplementary Fig. 4f, lower panel). Because they resulted from cross-contamination between adjacent AOIs, genes like *FN1* were removed from Tumor DEGs. For similar reasons, genes like *COL1A1* with low expression in *KRT14*$^{pos}$ clusters and high expression in *KRT14*$^{neg}$ clusters based on scRNA-seq were excluded from Tumor DEGs because of possible contamination (Supplementary Fig. 4f). Additionally, genes with ubiquitous expression based on scRNA-seq data were removed. Altogether, we obtained four

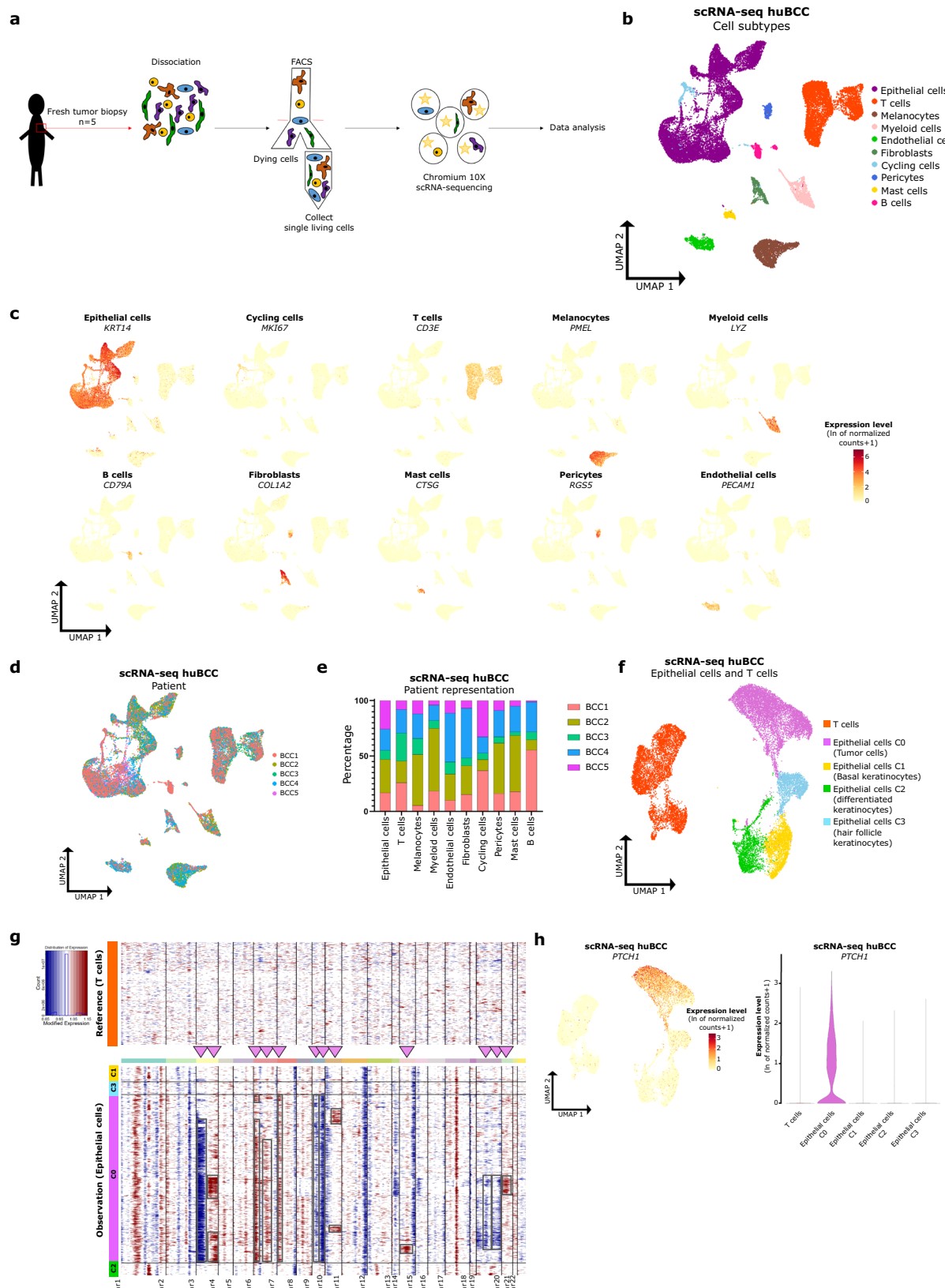

filtered signatures (Supplementary Fig. 4c; Supplementary Data 3): a spatial Tumor DEG signature (57 genes) composed of a nodular (spatial $T^{NOD}$, 45 genes) and an infiltrative (spatial $T^{INF}$, 12 genes) signatures; and a spatial Stroma DEG signature (46 genes) composed of a nodular (spatial $S^{NOD}$, 3 genes) and an infiltrative (spatial $S^{INF}$, 43 genes) signatures. As expected, the spatial Tumor DEG signature was enriched in the tumor compared to stroma AOIs (Fig. 2d). Conversely, the spatial Stroma DEG signature was enriched in the stroma compared to tumor AOIs (Fig. 2e). Consistently, when applied to tumor AOIs, spatial Tumor DEG signature segregated nodular from infiltrative AOIs on PCA plot (Fig. 2f). Similarly, when applied to stroma AOIs, spatial Stroma DEG signature segregated nodular from infiltrative AOIs (Fig. 2g).

**Fig. 1 | ScRNA-seq resolves the heterogeneous cell composition of infiltrative BCCs. a** Schematic of the single cell isolation from tumor excision to data analysis. **b** UMAP plot of 28,810 single cells integrated from five infiltrative BCCs colored according to ten distinct cell types that were annotated using canonical cell type markers (see **c**). **c** Expression level of canonical cell type markers in single cells of five huBCC samples represented as a color scale overlaid on the UMAP plots. **d** UMAP plot colored according to huBCC samples. **e** Bar graph illustrating the contribution by sample to each cell type subpopulation. **f** UMAP plot of $KRT14^{pos}$ epithelial and T cell subpopulations integrated from five infiltrative BCCs and colored according to subclustering used for the inferCNV

analysis. **g** Representative CNV heatmap from inferCNV analysis of epithelial cell clusters compared to T cells as reference (BCC1). Arrowheads and rectangles highlight inferred CNVs, after exclusion of gene expression changes shared by the $KRT14^{pos}$ epithelial cells from the five BCC samples. **h** Expression level of $PTCH1$ (BCC marker) in epithelial cells and T cells of five huBCC samples, represented as a color scale overlaid on the UMAP plot and as a violin plot per cell cluster. FACS fluorescence-activated cell sorting, scRNA-seq single-cell RNA sequencing, huBCC human basal cell carcinoma, UMAP uniform manifold approximation and projection, BCC basal cell carcinoma, C cluster, chr chromosome.

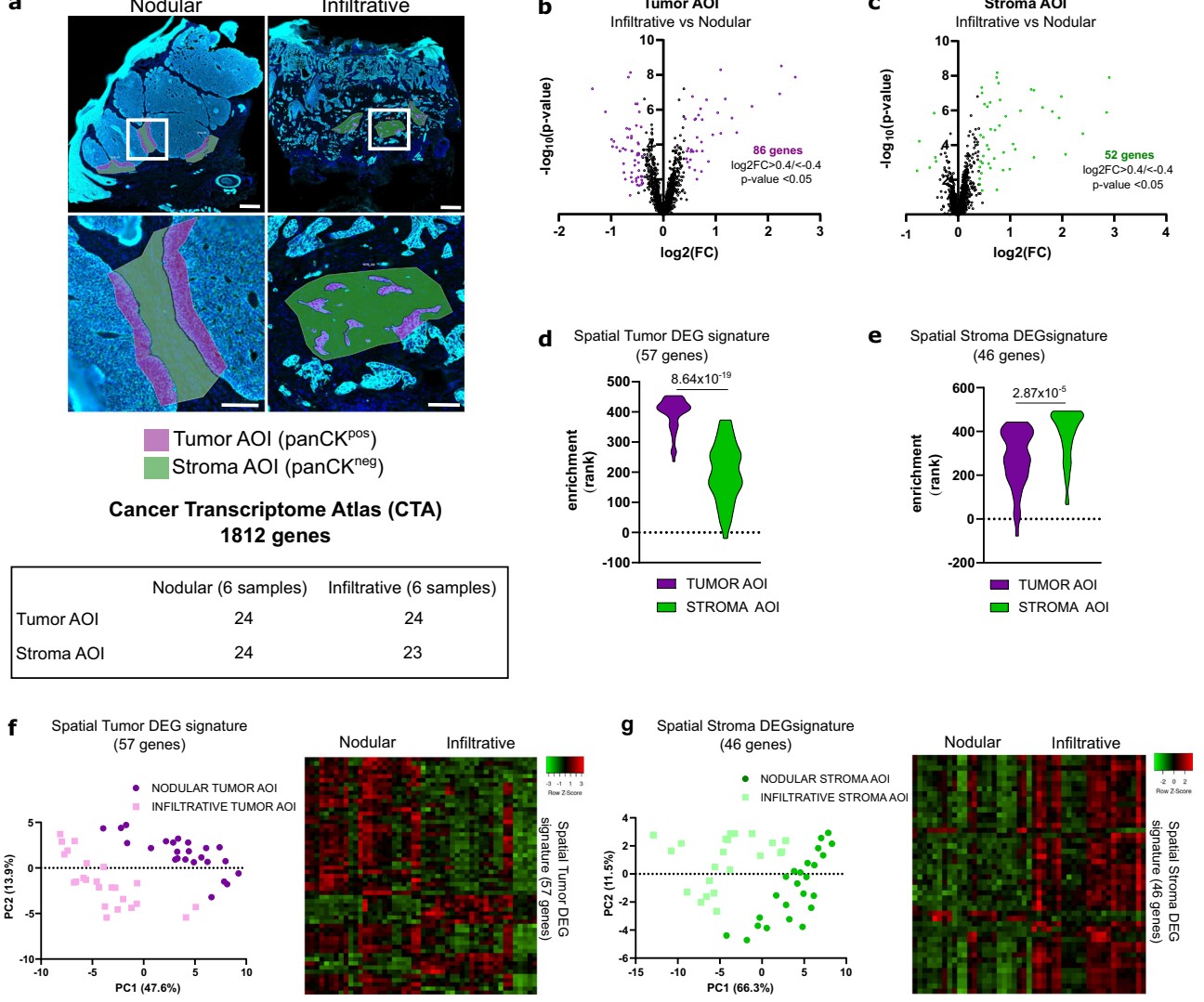

**Fig. 2 | Spatial transcriptomics defines compartment-specific signatures for nodular and infiltrative tumor–stroma interfaces. a** Immunofluorescence images of nodular and infiltrative BCC samples stained with DAPI and a panCK antibody showing the ROIs selection process and AOIs subdivision based on panCK staining. The table indicates the number of AOI for each tumor subtype (6 nodular and 6 infiltrative samples) and compartment (Tumor or Stroma). Scale bars indicate 300 μm (upper panels) and 100 μm (lower panels). **b** Volcano plot of the DEGs comparing the infiltrative to the nodular tumor AOIs. Eighty-six genes (highlighted in purple) show a log₂ fold change (log2FC) >0.4 or <−0.4 with a $p$ value <0.05. **c** Volcano plot of the DEGs comparing the infiltrative to the nodular stroma AOIs.

Fifty-two genes (highlighted in green) show a log2FC > 0.4 or <−0.4 with a $p$ value <0.05. **d** Violin plot of the spatial Tumor DEG signature (57 genes) enrichment in tumor and stroma AOIs. **e** Violin plot of the spatial Stroma DEG signature (46 genes) enrichment in tumor and stroma AOIs. **f** PCA plot and heatmap of the enrichment for the spatial Tumor DEG signature in individual tumor AOI. **g** PCA plot and heatmap of the enrichment for the spatial Stroma DEG signature in individual stroma AOI. $p$ Values in **d**, **e** were calculated using unpaired two-sided Student's $t$ test. Tumor and Stroma AOIs in **a**, **d**, **e** are depicted in purple and green, respectively. AOI area of interest, panCK pan-cytokeratin, log2FC log2 fold change, DEG differentially expressed genes, PC principal component, PCA principal component analysis.

Furthermore, individual $T^{NOD}$, $T^{INF}$, $S^{NOD}$, and $S^{INF}$ signatures showed specific enrichment in $T^{NOD}$, $T^{INF}$, $S^{NOD}$, and $S^{INF}$ AOIs, respectively (Supplementary Fig. 4g). Intriguingly, $S^{NOD}$ signature showed significant specificity for $T^{NOD}$ AOI as well, suggesting potential shared

transcriptional programs between adjacent compartments[28]. Taken together, these data show how DSP enabled the establishment of Tumor and Stroma interface signatures, specifically for nodular and infiltrative morphologies.

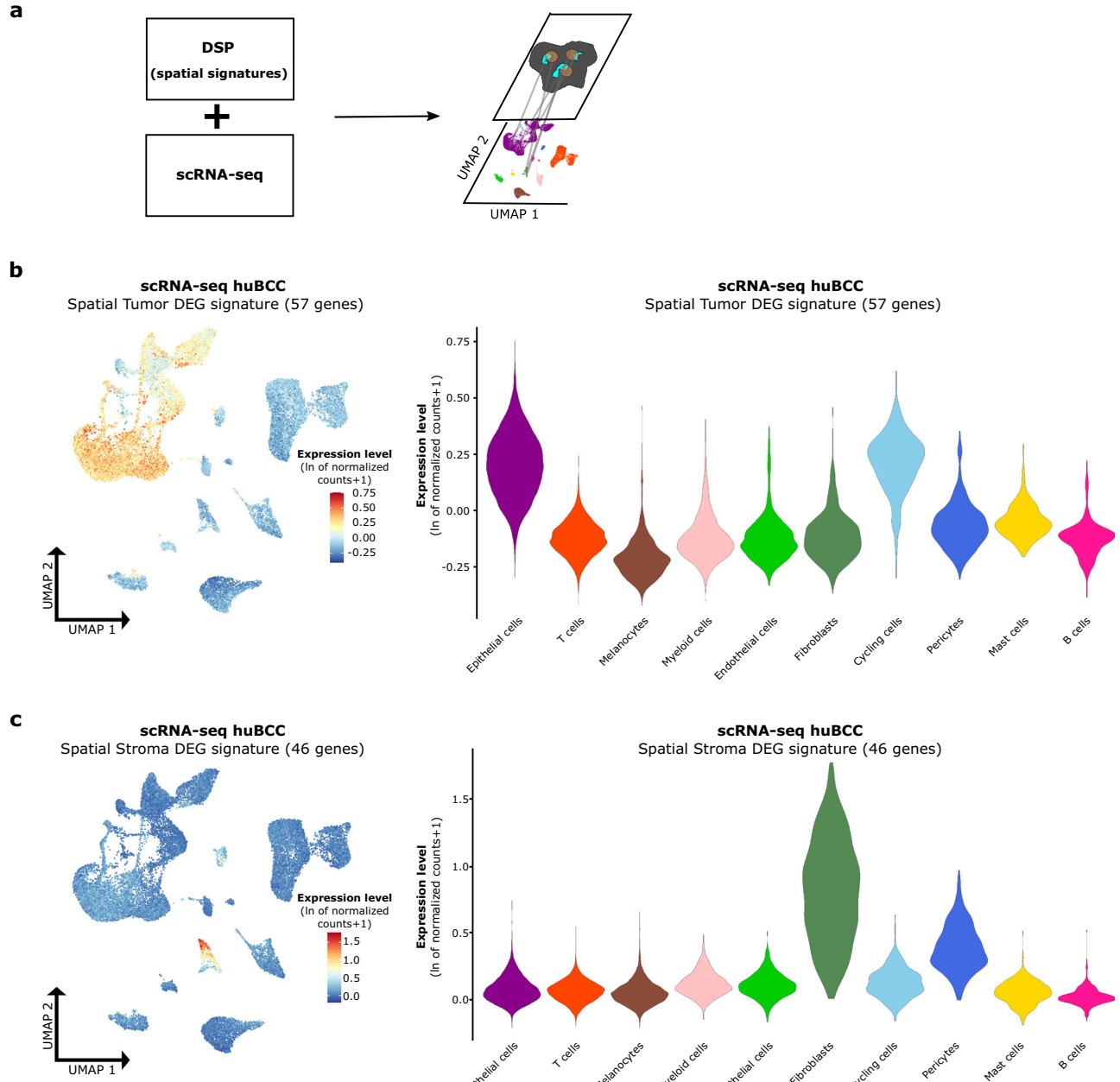

**Fig. 3 | Spatial signatures map cell subtypes composing the tumor–stroma interface in infiltrative BCCs. a** Overview of the analysis pipeline for the application of spatial signatures to scRNA-seq data. In brief, spatial Tumor and Stroma DEG signatures established with DSP were applied to the scRNA-seq dataset to enable the localization of cells with physical proximity on the UMAP plot. **b** Average expression level of the spatial Tumor DEG signature in single cells of five huBCC samples, represented as a color scale overlaid on the UMAP (left panel) and as violin plots per cell type (right panel). **c** Average expression level of the spatial Stroma DEG signature in single cells of five huBCC samples, represented as a color scale overlaid on the UMAP plot (left panel) and as violin plots per cell type (right panel). DSP digital spatial profiling, scRNA-seq single-cell RNA sequencing, UMAP uniform manifold approximation and projection, huBCC human basal cell carcinoma, DEG differentially expressed genes.

## Spatial signatures map cell subtypes composing the tumor–stroma interface in infiltrative BCCs

Having defined spatial tumor and stroma interface signatures, we thought to deconvolute the gene expression profiles into the cellular subpopulation obtained by scRNA-seq (Fig. 3a), and the apparent complexity of their potential interactions. To identify cell subpopulations in physical proximity at the tumor–stroma interface, we applied DSP-defined spatial Tumor and Stroma DEG signatures on scRNA-seq UMAP of infiltrative BCCs. Consistently, the spatial Tumor DEG signature (57 genes) showed the highest enrichment in the previously identified tumor cell cluster (Fig. 3b and Supplementary Fig. 5). The

spatial Stroma DEG signature (46 genes) demonstrated the highest enrichment in the fibroblast cluster (Fig. 3c), previously reported to drive BCC infiltration[31]. Collectively, by integrating DSP and scRNA-seq technologies, our data identify the cellular contributors to the tumor-stroma interface in infiltrative BCCs.

## Tumor cells within the invasive niche harbor epithelial collective migration features

To further dissect the mechanisms driving tumor invasion, we next thought to interrogate the transcriptional program of tumor cells found within the invasive niche, taking advantage of tumor cell

heterogeneity observed by scRNA-seq. To do so, we subclustered the tumor cell subpopulation previously identified and obtained 15 unsupervised clusters with satisfactory patient integration and gene expression complexity (Fig. 4a and Supplementary Fig. 6). We then computed the difference between spatial $T^{INF}$ and $T^{NOD}$ signature enrichment scores for individual tumor cells (Fig. 4b). Clusters were ranked based on their mean difference between spatial $T^{INF}$ and $T^{NOD}$ signature enrichment scores, and the 15 ranked clusters divided into 3 groups according to their infiltrative profile: Low (TC1–4), Med (TC5–11), and High (TC12–15) (Fig. 4c, d). As expected from the nodular to infiltrative histological progression BCCs typically manifest in patients, we observed a negative correlation between spatial $T^{NOD}$ and $T^{INF}$ enrichments (Supplementary Fig. 7a), with progressive enrichment for spatial $T^{INF}$ signature from TC1 to TC15 (Supplementary Fig. 7b, upper panels, 7c), and reversed enrichment for spatial $T^{NOD}$ signature from TC15 to TC1 (Supplementary Fig. 7b, lower panels, 7c). Thus, to identify the transcriptional reprogramming underlying nodular to infiltrative progression, we looked at the differentially expressed genes between Low and High infiltrative clusters (Supplementary Fig. 7d, Supplementary Data 4). Intriguingly, gene set enrichment analysis of the significantly up-regulated genes highlighted skin differentiation-related biological processes (epithelial cell differentiation, cell differentiation) (Supplementary Fig. 7e, f). Consistently, the enrichment of a basal signature[21], KEGG Hedgehog signaling signature, and KEGG BCC signature were globally higher in the Low (TC1–4) compared to the High (TC12–15) infiltrative clusters (Fig. 4e, upper panels, Supplementary Fig. 7g). In contrast, epidermal differentiation signature[21] enrichment was globally higher in the High (TC12–15) compared to the Low (TC1–4) infiltrative clusters (Fig. 4e, lower panels). The presence of differentiation features in the High infiltrative clusters (TC12–15) raised the hypothesis of epithelial collective migration depending on cell-cell adhesion complexes[32,33] for BCC invasion. Consistently, High (TC12–15) infiltrative clusters demonstrated a higher enrichment for the collective migration signature[34] compared to the other clusters (Fig. 4f, upper panels). In contrast, the epithelial-to-mesenchymal transition (EMT) signature did not show significant preferential enrichment (Fig. 4f, lower panels).

Because we observed a gradual and inversed enrichment of $T^{NOD}$ and $T^{INF}$ signatures in scRNA-seq, we took advantage of the pseudotime analysis to perform an unsupervised ordering of the cells along a path according to their transcriptional profile[35]. Remarkably, when starting from the Low infiltrative clusters, the pseudotemporal trajectory progressed to Med infiltrative and ended with High infiltrative clusters (Fig. 4g, upper panel), as illustrated by the progressive decrease in expression of $T^{NOD}$ signature genes (*MPPED1*, *PTCH1*) alongside a progressive increase in expression of $T^{INF}$ signature genes (*KRT6A*, *SFN*) (Fig. 4g, lower panels).

Importantly, we confirmed the negative correlation of the spatial $T^{NOD}$ marker *MPPED1* and the spatial $T^{INF}$ marker *KRT6A* expression at the tumor-stroma interface by RNA FISH (Supplementary Fig. 8a). Consistently with the pseudotemporal trajectory, the progression of *KRT6A*^low^/*MPPED1*^high^ to *KRT6A*^high^/*MPPED1*^low^ staining intensities paralleled Low-infiltrative to High-infiltrative morphological progression (Supplementary Fig. 8b, c). Along this pseudotemporal trajectory mirroring the histopathological progression seen in BCCs, we confirmed a progressive reduction in the expression of genes that are specifically expressed in basal epidermal keratinocytes (*DST*, *COL17A1*) (Fig. 4h), while a progressive increase in the expression of differentiation-specific genes (*KRT1*, *CALML5*) as well as epidermal cohesion genes (*CLDN4*, *NECTIN1*, *DSG3*) supporting collective migration (Fig. 4i). Previous studies revealed higher expression of ECM, ECM receptor and ECM-remodeling genes like *FN1*[13,31], *POSTN*[13], *CCN4*[13], *COL3A1*[13], *ADAMTS2*[13], *LRRC15*[13], *ITGA5*[31], *ITGAV*[36], and *ITGB6*[36] in infiltrative compared to nodular BCCs, questioning their tumoral expression and potential EMT. Consistent with epithelial collective migration

rather than EMT, these genes were barely or not expressed in the tumor cell subpopulations (Supplementary Fig. 9), suggesting their predominant stromal expression.

Altogether, spatial signatures mapped tumor cells composing the invasive niche in scRNA-seq data, revealing how their transcriptional reprogramming supports collective migration.

## CAFs found in the invasive niche harbor ECM-remodeling features

Collective migration requires adjacent stromal fibroblasts to support invasion[37]. We previously identified fibroblasts as preferential spatial interactors at the tumor-stroma interface of infiltrative BCCs (Fig. 3c). We thus thought to further dissect the transcriptional reprogramming of fibroblasts found within the invasive niche, taking advantage of their heterogeneity on scRNA-seq UMAP (Fig. 5a). To do so, we subclustered the fibroblast cell subpopulations and obtained 4 unsupervised clusters with satisfactory patient integration and gene expression complexity (Figs. 3c and 5a and Supplementary Fig. 10). We then computed the difference between spatial $S^{INF}$ and $S^{NOD}$ signature enrichment scores for individual fibroblasts (Fig. 5b). Clusters were ranked based on their mean difference between spatial $S^{INF}$ and $S^{NOD}$ signatures enrichment scores: Low (FC1), Med low (FC2), Med high (FC3), and High (FC4) (Fig. 5c, d). As expected, we observed a negative correlation between spatial $S^{NOD}$ and $S^{INF}$ enrichments (Supplementary Fig. 11a), with progressive enrichment for the spatial $S^{INF}$ signature from Low (FC1) to High (FC4) infiltrative fibroblast clusters (Supplementary Fig. 11b, upper panels, 11c), and reversed enrichment for the spatial $S^{NOD}$ signature from High (FC4) to Low (FC1) infiltrative fibroblast clusters (Supplementary Fig. 11b, lower panels, 11c). Activated fibroblasts located in close proximity to the tumor are called cancer-associated fibroblasts (CAFs). They express vimentin, and a large subset of CAFs also expresses high levels of alpha smooth muscle actin (αSMA)[38,39]. In contrast, quiescent fibroblasts are non-contractile, spindle-shaped, and in a resting state[38,39]. Whereas quiescent fibroblasts[40] were identified in FC3 (Fig. 5e, upper panels), CAF signature markers identified by scRNA-seq in several studies[41] showed the highest enrichment in both Low (FC1) and High (FC4) infiltrative fibroblast clusters (Fig. 5e, lower panels), although with different profiles (Fig. 5f). Indeed, when looking at the significantly up-regulated genes in High (FC4) versus Low (FC1) infiltrative fibroblast clusters (Supplementary Fig. 11d, e and Supplementary Data 5), gene set enrichment analysis highlighted ECM-remodeling-related biological processes (extracellular matrix organization, extracellular structure organization) (Supplementary Fig. 11f). The identified ECM-remodeling profile of High infiltrative CAFs (FC4) was further confirmed by their enrichment for the KEGG regulation of actin cytoskeleton and the KEGG ECM-receptor interaction signatures (Fig. 5g). We also confirmed the stromal fibroblast origin of the previously reported infiltrative BCC-specific markers (*FN1*[13,31], *POSTN*[13], *CCN4*[13], *COL3A1*[13], *ADAMTS2*[13], *LRRC15*[13]), with specific enrichment in CAFs found within the invasive niche (Supplementary Fig. 11g). Importantly, analysis for inferred copy number variations showed significantly lower rate of chromosomal aberrations in the fibroblast compared to the tumor cell clusters (Supplementary Fig. 11h), arguing against tumor epithelial transdifferentiation upon EMT into CAF-like populations[42]. Altogether, spatial signatures mapped fibroblasts composing the invasive niche in scRNA-seq, revealing how their prominent ECM-remodeling features may support tumor cell collective migration.

## *INHBA* is preferentially expressed in the highly infiltrative tumor tips

We next inferred from pseudotime analysis the various genes driving tumor cell progression trajectory (Supplementary Data 6) and overlapped the identified genes with a series of known secreted CAF activators (transforming growth factor-β (TGF-β), other TGF-β superfamily

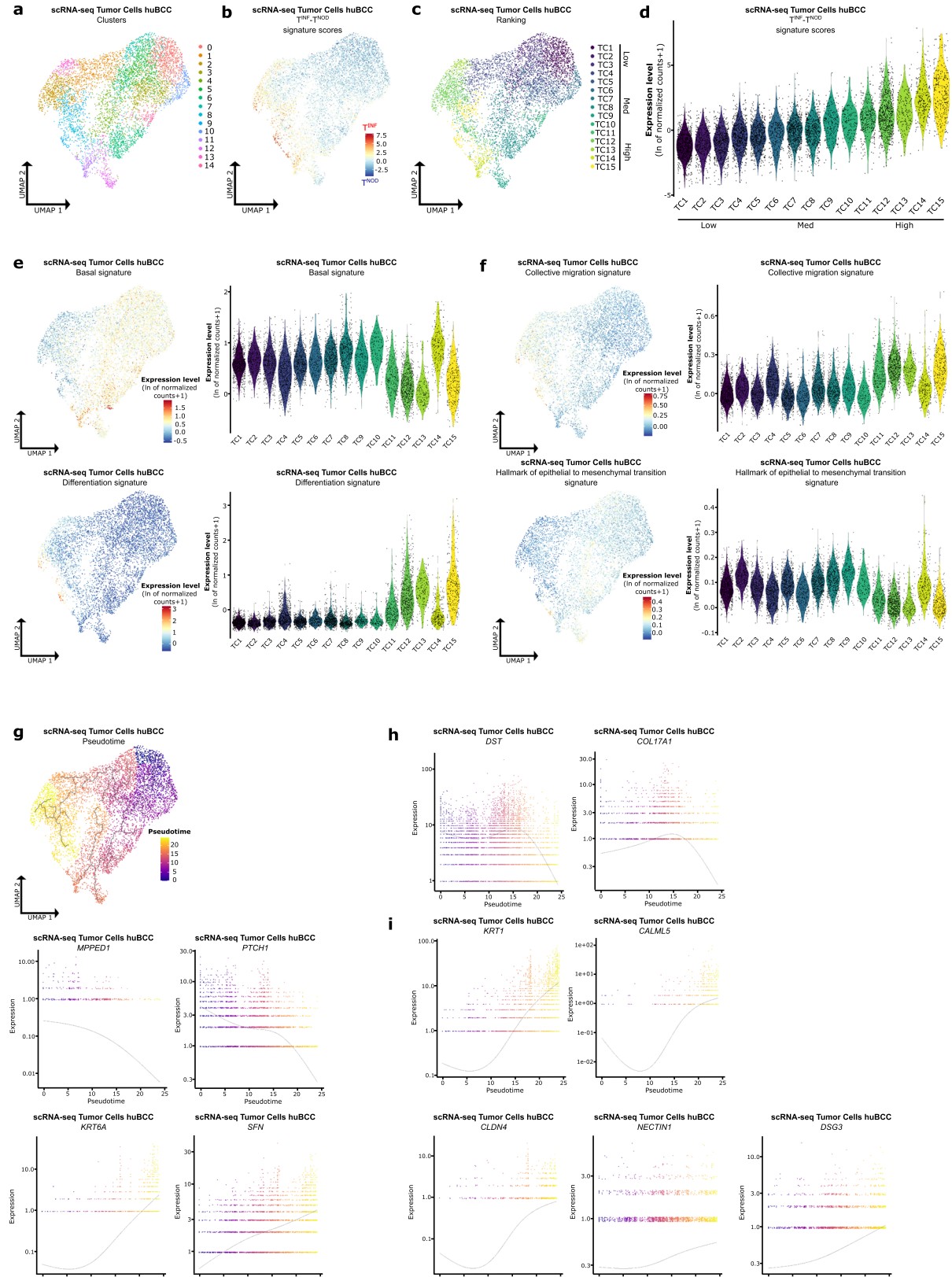

members, platelet-derived growth factors (PDGFs), epidermal growth factor (EGF) family members, fibroblast growth factors (FGFs) and sonic hedgehog (SHH))[38] (Fig. 6a). Although expressed in infiltrative tumor cells, *TGFB1*, previously reported to drive BCC infiltration[31], did not show significant autocorrelation with the pseudotime trajectory (Supplementary Fig. 12a). By contrast, *INHBA*, coding for the inhibin

beta A subunit of other TGF-β superfamily members, including the homodimer Activin A, the heterodimer activin AB or their antagonist Inhibin A (a heterodimer of INHBA and an α subunit)[43], showed the highest pseudotime autocorrelation and very specific association with Highly infiltrative tumor clusters (TC12–15) (Fig. 6a, b). The related *INHBB* was expressed at lower levels and more homogeneously

**Fig. 4 | Tumor cells within the invasive niche harbor epithelial collective migration features. a** Unsupervised clustering of the tumor cell subpopulations represented as a UMAP plot. **b** Average expression level of the difference between the spatial T$^{INF}$ and spatial T$^{NOD}$ signature scores in tumor cell subpopulations of the scRNA-seq dataset, represented as a color scale overlaid on the UMAP plot. **c** UMAP illustrating the unsupervised clusters ranked according to their mean difference between the spatial T$^{INF}$ and spatial T$^{NOD}$ signature scores in tumor cell subpopulations of the scRNA-seq dataset. **d** Average expression level of the difference between the spatial T$^{INF}$ and spatial T$^{NOD}$ signature scores in tumor cell subpopulations of the scRNA-seq dataset, represented as violin plots per ranked cluster. **e** Average expression of the "basal" and "differentiation" signatures in tumor cell subpopulations of the scRNA-seq dataset, represented as a color scale overlaid on the UMAP plot (left panels) and as violin plots per ranked cluster. **f** Average expression level of the "collective migration" and "Hallmark of epithelial to mesenchymal transition" signatures in tumor cell subpopulations of the scRNA-seq dataset, represented as a color scale overlaid on the UMAP plot (left panels) and as violin plots per ranked cluster (right panels). **g** Pseudotemporal trajectory across tumor cell subpopulations, starting point in Low infiltrative cluster TC1 (upper panel). Expression of spatial T$^{NOD}$ (*MPPED1*, *PTCH1*) and spatial T$^{INF}$ (*KRT6A*, *SFN*) markers along the pseudotemporal trajectory of tumor cell subpopulations (lower panels). **h** Expression of basal (*DST*, *COL17A1*) markers along the pseudotemporal trajectory of tumor cell subpopulations. **i** Expression of differentiation (*KRT1*, *CALML5*), tight junction (*CLDN4*), adherens junction (*NECTIN1*), and desmosomal junction (*DSG3*) markers along the pseudotemporal trajectory of the tumor cell subpopulations. scRNA-seq single-cell RNA sequencing, huBCC human basal cell carcinoma, UMAP uniform manifold approximation and projection, T$^{INF}$ infiltrative tumor, T$^{NOD}$ nodular tumor, TC tumor cluster.

(Supplementary Fig. 12b). Expression of *INHA*, coding for the α-subunit of Inhibin, and *FST*, coding for the secreted activin antagonist follistatin, was also very low (Supplementary Fig. 12b), suggesting that the increased levels of *INHBA* mainly give rise to homodimeric Activin A. To consolidate these data, we interrogated tumor AOIs for *INHBA* and *INHBB* expression (as part of the CTA panel). *INHBA* expression was significantly enriched in infiltrative tumor AOIs, while *INHBB* expression was significantly decreased compared to nodular tumor AOIs (Supplementary Fig. 12c). Using RNA FISH, we found consistent increased expression of tumor *INHBA* at the tumor–stroma interface of infiltrative compared to nodular BCCs (Fig. 6c, d), particularly in tumor areas with High infiltrative morphology (Fig. 6d). In contrast, *TGFB1* showed a more homogenous expression pattern (Supplementary Fig. 12d). Importantly, we confirmed the tumor origin of *INHBA* expression at the invasive tip (Fig. 6e). As expected from the combination of the spatial and scRNA sequencing data, we found positive correlations between the expression of *INHBA* and collective migration marker genes (*CLDN4*, *NECTIN1*, and *DSG3*) at the tumor-stroma interface (Supplementary Fig. 13). Taken together, these results highlight the preferential expression of *INHBA* in tumor cells located at the invasive tip of infiltrative BCCs.

### Activin A signals between spatially-connected infiltrative tumor cells and associated CAFs

Having identified Activin A as a potential signaling molecule that is released by infiltrative tumor cells, we next checked whether Activin A may influence the identified ECM-remodeling CAFs. Therefore, we first checked the expression of Activin A receptors. Activin A typically signals through ACVRII/ACVRIB (ALK4) heterodimers[44,45]. Indeed, skin fibroblast and skin CAF responsiveness to Activin A was previously shown to be mediated at least in part by ALK4[46–48]. Here, we confirmed *ACVRIIA/B* and *ACVRIB* (ALK4) expression in BCC CAFs (Supplementary Fig. 14). To confirm downstream activation, we applied a signature of CAFs exposed to Activin A previously established in ref. 46. The signature of Activin A-exposure was highly enriched in ECM-remodeling CAFs found in the invasive niche (FC4) (Fig. 7a). Consistently, tumor cell expression of *INHBA* positively correlated with the expression of Activin A-exposure genes in the adjacent stroma, based on DSP (Supplementary Fig. 15a). To demonstrate the physical proximity between tumor cells expressing *INHBA* and ECM-remodeling CAFs, we co-stained nodular and infiltrative BCCs for *INHBA* and Activin A-exposure signature genes like *FN1* and *POSTN* using RNA FISH. Remarkably, we found a positive correlation between *INHBA* expression in tumor cells and the surrounding expression of Activin A-exposure genes (*FN1* or *POSTN*) in CAFs (Fig. 7b and Supplementary Fig. 15b), with preferential expression in tumor areas with High infiltrative morphology (Fig. 7c and Supplementary Fig. 15c). Of note, out of the previously identified major contributors of infiltrative BCCs, *FN1*, *POSTN*, *COL3A1*, and *LRRC15* were identified as Activin A target genes[46,47]. Altogether, these expression patterns strongly suggest that the INHBA homodimer

Activin A, through paracrine signaling, governs tumor cell and CAF interactions in the invasive niche (Fig. 7d).

Overall, using integrated spatial and scRNA-seq, we deciphered the cellular constituents, their transcriptional reprogramming, and potential crosstalk within the invasive niche of aggressive skin tumors. Importantly, we highlight a continuous progression in the infiltrative properties of tumors supported by adjacent ECM-remodeling CAFs, ending up in the spatially organized invasive niche, where Activin A signaling may serve for reciprocal intercellular crosstalk.

## Discussion

We combined scRNA-seq and spatial transcriptomics to decipher the complex biological organization of the invasive front in aggressive BCCs. To do so, we integrated 28,810 single cell transcriptomes with spatially-resolved transcriptomes of 48 tumor areas and 47 adjacent stroma areas in infiltrative and non-infiltrative BCCs. Among the various cell types composing infiltrative tumors, we identified a committed tumor cell subpopulation with collective migration features and ECM-remodeling CAFs as being the principal contributors to the invasive niche. Using differential gene expression and pseudotime analysis, we showed that Activin A paracrine signaling governs their reciprocal transcriptional reprogramming. Thus, integration of spatial and single-cell transcriptomics can resolve the cellular and molecular complexity of tumor invasion to provide unique mechanistic insights.

scRNA-seq captures the transcriptional activity of individual cells in tissues but lacks spatial information. By contrast, spatial transcriptomics provides transcriptional activities in spatially defined tissue areas but does not provide information about individual cells. By integrating single-cell and spatial RNA-seq, we may now map the transcriptional profile of single cells to their native tissue environment[49], to foster our understanding of tumor biology[21,50–52]. In contrast with these previous publications, we used here an alternative approach, where a high number of spatially-restricted areas of interest in tumors are defined transcriptionally, integrated, and then mapped onto scRNA-seq data to obtain cellular resolution. Our integrative approach was also used to filter data containing RNA cross-contaminations between adjacent AOIs[30]. Depending on the size of the capture spots and the gene library available, spatial resolution may improve in the future. Nevertheless, our filtered spatial signatures, when used on scRNA-seq data, identified transcriptionally-characterized clusters in physical proximity at the leading edge of infiltrative tumors. Importantly, the data may serve as a resource to interrogate additional tumor types or compare leading edge signatures and composition.

Previous studies using bulk RNA transcriptomics reported the expression of mesenchymal ECM and ECM-remodeling genes like *FN1*[13,31], *POSTN*[13], *CCN4*[13], *COL3A1*[13], *ADAMTS2*[13], and *LRRC15*[13] in infiltrative BCCs. These studies, however, lacked spatial resolution, questioning their expression by tumor cells undergoing EMT. Our integrated spatial and scRNA-seq confirmed the expression of these

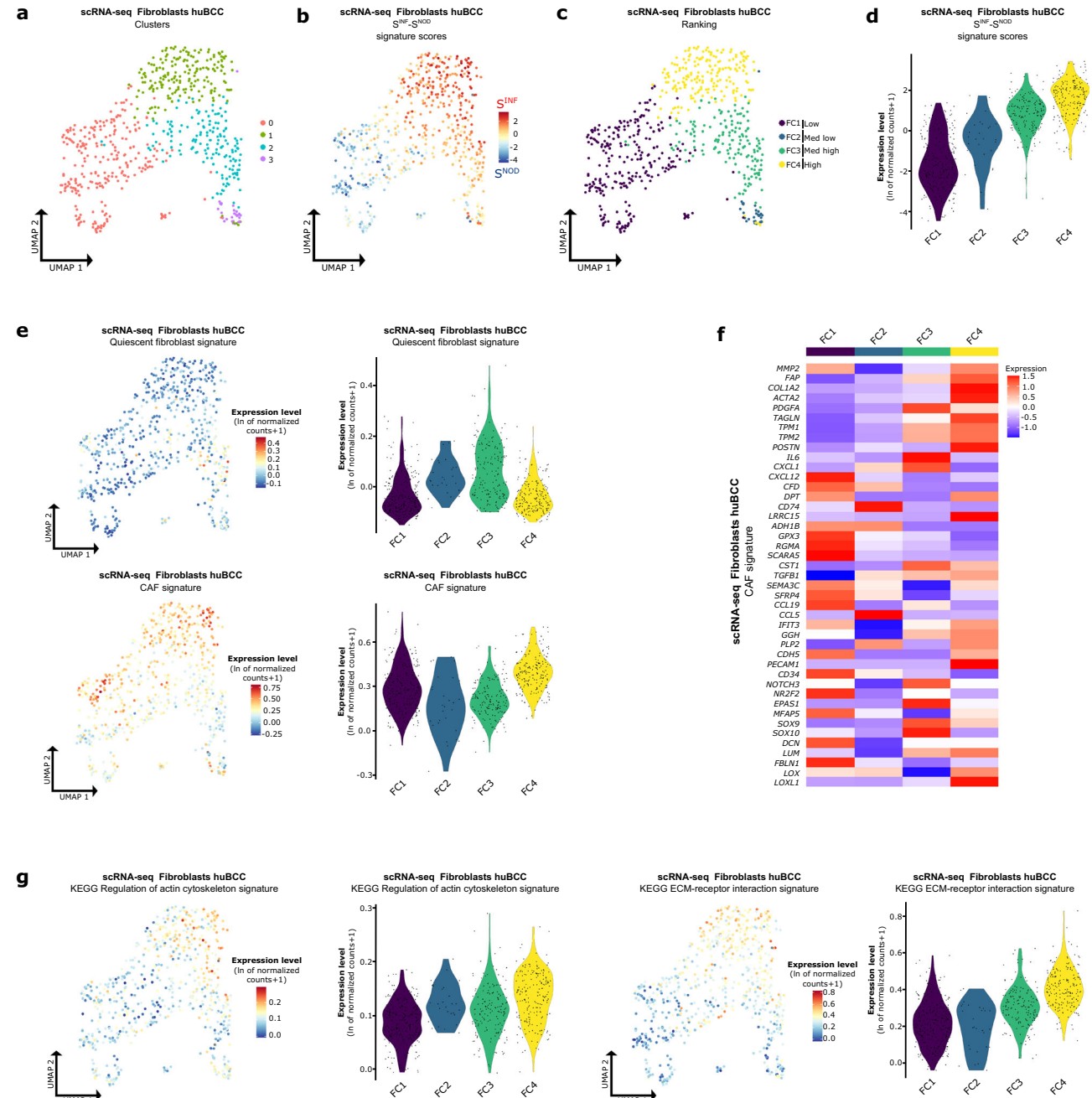

**Fig. 5 | CAFs found within the invasive niche harbor ECM-remodeling features. a** Unsupervised clustering of the fibroblast subpopulations represented as a UMAP plot. **b** Average expression level of the difference between spatial S$^{INF}$ and spatial S$^{NOD}$ signature scores in fibroblast subpopulations of the scRNA-seq dataset, represented as a color scale overlaid on the UMAP plot. **c** UMAP illustrating the unsupervised clusters ranked according to their mean difference between the spatial S$^{INF}$ and spatial S$^{NOD}$ signature scores in fibroblast subpopulations of the scRNA-seq dataset. **d** Average expression level of the difference between the spatial S$^{INF}$ and spatial S$^{NOD}$ signature scores in fibroblast subpopulations of the scRNA-seq dataset, represented as violin plots per ranked cluster. **e** Average expression level of the "Quiescent fibroblast" and "CAF" signatures in fibroblast subpopulations of the

scRNA-seq dataset, represented as a color scale overlaid on the UMAP plot (left panels) and as violin plots per ranked cluster (right panels). **f** Heatmap of the average expression level of genes present in the "CAF" signature in fibroblast subpopulations of the scRNA-seq dataset. **g** Average expression level of the "KEGG regulation of actin cytoskeleton" (left panels) and "KEGG ECM-receptor interaction" (right panels) signatures in fibroblast subpopulations of the scRNA-seq dataset, represented as a color scale overlaid on the UMAP plot and as violin plots per ranked cluster. scRNA-seq single-cell RNA sequencing, huBCC human basal cell carcinoma, UMAP uniform manifold approximation and projection, S$^{INF}$ infiltrative stroma, S$^{NOD}$ nodular stroma, FC fibroblast cluster, CAF cancer-associated fibroblasts, ECM extracellular matrix.

---

genes in infiltrative tumors, but predominantly by CAFs and not tumor cells. In the contrary, transcriptional reprogramming in infiltrative tumor cells mostly involved genes implicated in epithelial commitment and collective migration, which typically relies on epithelial cell-cell interactions based on tight and adherens junctions. Ji et al. previously reported a tumor-specific population with EMT features at

the leading edge in the proximity of CAFs in SCC[21], harboring specific *INHBA* expression. While it supports a common role for Activin A in governing the invasive niche of epithelial skin cancers, it questions the connection between *INHBA* expression and tumor migration mode. Of note, our study highlights the transcriptional heterogeneity of *INHBA*$^{pos}$ cells seen within scRNA-seq data, but not captured by the DSP

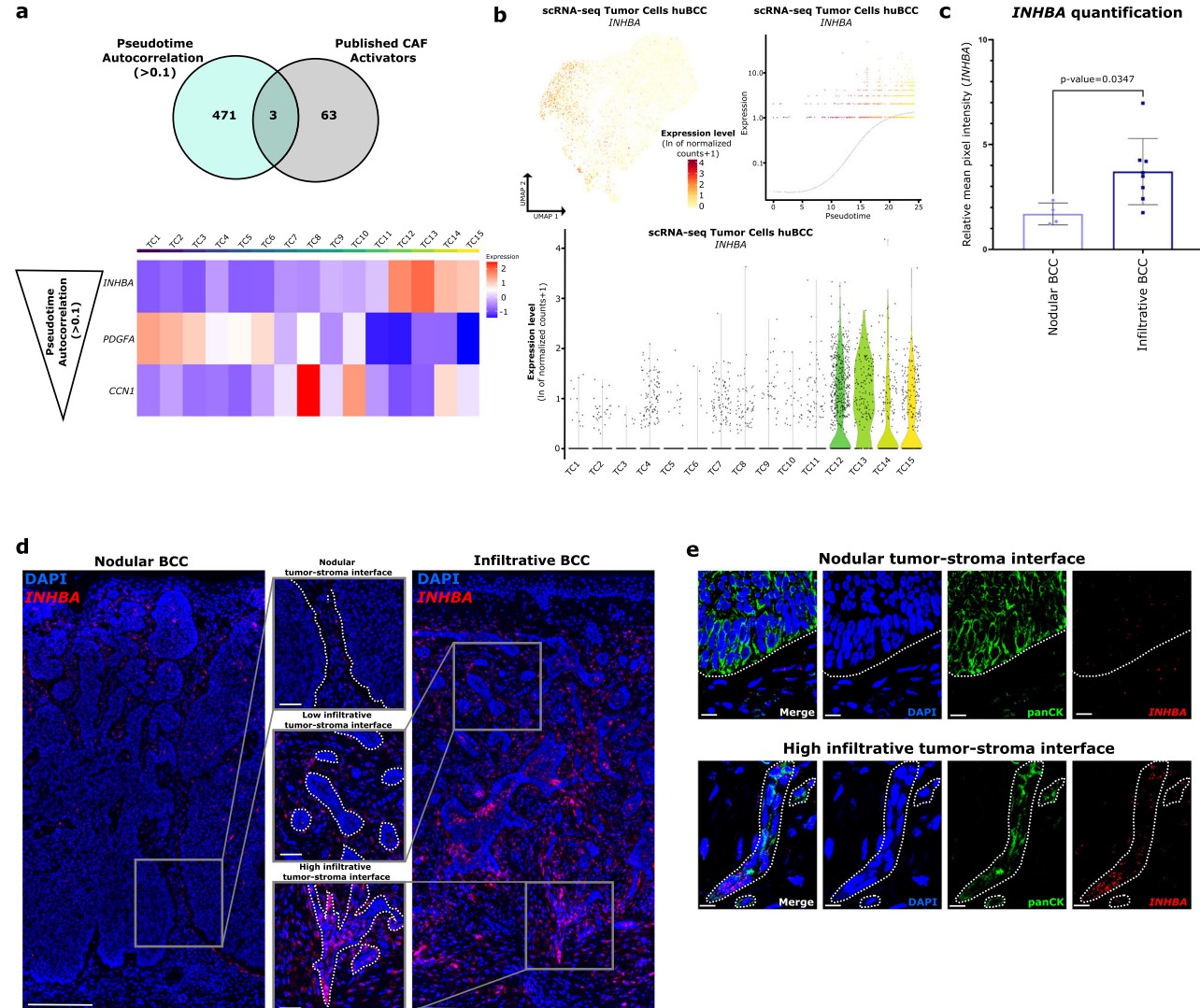

**Fig. 6 | *INHBA* is preferentially expressed in the highly infiltrative tumor tips.**
**a** Venn diagram showing the overlap between genes with a positive (>0.1) pseudotime autocorrelation index and known CAF activators (upper panel). Lower panel shows the average expression level per ranked cluster of the three overlapping genes, ordered by decreasing autocorrelation index, in tumor cell subpopulations of the scRNA-seq dataset. **b** Expression level of *INHBA* in the tumor cell subpopulations of the scRNA-seq dataset, represented as a color scale overlaid on the UMAP plot (upper left panel) and as violin plots per ranked cluster (lower panel). Expression level per cell of *INHBA* along the pseudotemporal trajectory of tumor cell subpopulations (upper right panel). **c** Bar graph representing the relative intensity of *INHBA* expression in tumor cells located at the tumor-stroma interface of nodular (*N* = 4) and infiltrative (*N* = 8) BCC samples (measured in *n* ≥ 15 regions/sample). Horizontal bars indicate the mean ± SD.

*p* Value was calculated by unpaired two-sided Student's *t* test. **d** Representative scan images of nodular and infiltrative BCC sections stained with DAPI (blue) and *INHBA* (red) probe for RNA FISH. Upper, intermediate and lower zoom areas show tumor-stroma interface with distinct Nodular, Low, and High infiltrative morphologies respectively. Dotted lines highlight the tumor-stroma interface. Scale bars indicate 200 µm (main panel) and 50 µm (zoom panels).
**e** Representative confocal images of Nodular and High infiltrative tumor-stroma interfaces stained with DAPI (blue), pan-cytokeratin antibody (panCK, green), and *INHBA* (red) probe for RNA FISH. Dotted lines highlight the tumor-stroma interface. Scale bars indicate 10 µm. CAF cancer-associated fibroblasts, TC tumor cluster, scRNA-seq single-cell RNA sequencing, huBCC human basal cell carcinoma, UMAP uniform manifold approximation and projection, BCC basal cell carcinoma. Source data for **c** are provided as a Source data file.

profiling. Despite predominant differentiation and collective migration features, a few *INHBA*^pos cells identified in our study may harbor discrete EMT features. Overall, it is suggestive of few, rare, partial EMT tumor cells governing collective migration during development and cancer progression[9,53] but requires additional studies with improved gene and spatial resolution to be confirmed.

Our data highlight how aggressive tumors repurpose tissue regeneration mechanisms to progress[54,55]. During tissue repair, keratinocytes stimulate fibroblasts, which in turn drive keratinocyte proliferation, migration, and differentiation to restore the epidermal barrier[32,56]. By analogy, spatial annotation of scRNA-seq data highlights the transcriptional reprogramming of committed, migrating tumor

cells and adjacent fibroblasts, supporting reciprocal biological crosstalk (Fig. 7d). Consistently, Activin A paracrine signaling, identified at the tumor leading edge of BCCs, is involved in tissue repair and fibrosis. This activity involves the activation of ECM-remodeling fibroblasts[45,57,58] and is associated with poor prognosis in various cancer types, including skin cancer[46,59–61]. While mechanisms may differ between cancer types, enhanced ECM stiffness induced by Activin A, which was observed in healing skin wounds[47], may drive tumor progression[62,63]. We previously reported the implication of TGF-β1, a key regulator of myofibroblast differentiation, in driving BCC infiltration[31]. Interestingly, *TGFB1* and *INHBA* display different expression patterns in infiltrative BCCs, suggesting that TGF-β1 and Activin A

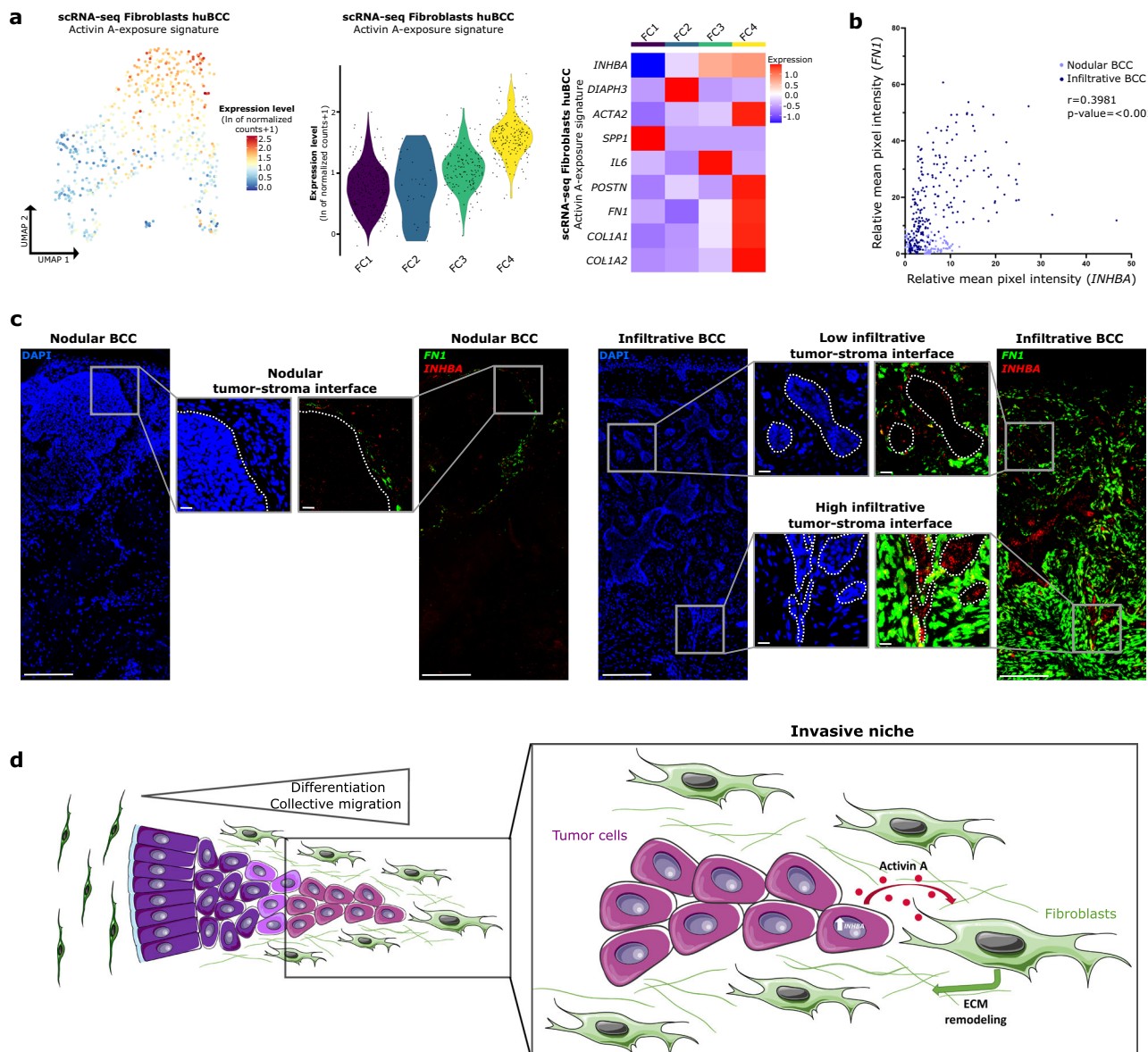

**Fig. 7 | Activin A signals between spatially-connected infiltrative tumor cells and associated CAFs. a** Average expression level of the signature resulting from Activin A exposure in the fibroblast subpopulations of the scRNA-seq dataset, represented as a color scale overlaid on the UMAP plot (left panel) and as violin plots per ranked cluster (middle panel). Right panel shows the heatmap of average expression level of genes present in the signature in ranked clusters of the fibroblast subpopulations. **b** Correlation analysis between tumor *INHBA* and adjacent stromal *FN1* relative intensities in nodular (*N* = 4) and infiltrative (*N* = 8) BCCs (*n* ≥ 10 regions/sample). *R* coefficient and *p* value were calculated by the Spearman correlation test and a two-sided statistical significance test, respectively. **c** Representative scan images of nodular and infiltrative BCC sections stained with DAPI (blue), *INHBA* (red), and *FN1* (green) probes for RNA FISH. Upper, intermediate and lower zoom areas show tumor-stroma interface with distinct Nodular, Low, and High infiltrative morphologies respectively. Dotted lines highlight the tumor-stroma interface. Scale bars indicate 200 μm (main panels) and 20 μm (zoom panels). **d** Summary scheme depicting the continuum of tumor infiltrative properties, with a zoom on the cellular and molecular interactions identified within the invasive niche. Image designed with Servier Medical Art (smart.servier.com). scRNA-seq single-cell RNA sequencing, huBCC human basal cell carcinoma, UMAP uniform manifold approximation and projection, FC fibroblast cluster, BCC basal cell carcinoma, ECM extracellular matrix. Source data for **b** are provided as a Source data file.

expression are regulated distinctly in tumors. While *TGFB1* is homogeneously expressed, *INHBA* expression is restricted to highly infiltrative tumor tips. TGF-β1 inhibitors, including TGFβRI kinase inhibitors with ALK4 blocking activity, failed to deliver anticipated results in clinical trials involving patients with different types of cancer[64]. However, their limited clinical success was not related to the ALK4 blocking activity. Currently tested Activin A/activin receptors inhibitors, including recombinant follistatin (FST), neutralizing Activin A antibodies and soluble activin receptors, target Activin A without affecting TGFβR1. As such, Activin A signaling-targeting strategies

(tested in phase I/IIa clinical trials for muscular dystrophy[65]) may differ in terms of adverse effects and efficacy and offer an attractive alternative to TGF-β1 inhibitors.

Altogether, our data show how tumor cell differentiation, collective migration, and ECM-remodeling CAFs are spatially and transcriptionally connected. Our data support that Activin A-mediated collective migration and matrix remodeling required for tissue repair are repurposed by committed skin tumor cells to invade surrounding tissues, and may be specifically targeted in the clinics[45]. While the limited sample cohort in our study may not reflect the potential

heterogeneity of invasion modes in BCC, our findings demonstrate the power of integrated single-cell and spatial transcriptomics to decipher the spatial organization and intercellular signals governing the invasive niche of skin cancer.

## Methods

### Human samples

Studies were approved by the institutional review board of Lausanne University Hospital CHUV, and the local ethics committee, in accordance with the Helsinki Declaration (CER-VD 2020-02204). Written informed consent was obtained from each patient. Based on the principle of voluntary participation, no compensation was provided to the patients in this study. Samples from scRNA-seq were collected from Mohs specimen residual debulking piece and processed immediately after surgical resection. Samples for DSP and RNA FISH were obtained from formalin-fixed paraffin-embedded tumor tissues, from respectively different patients. Histological diagnoses were obtained from an independent dermatopathologist. None of the patients were immunosuppressed. Comprehensive information on age, gender, tumor localization, and previous treatment are listed in Supplementary Fig. 1a.

### Tissue dissociation and single-cell sorting

Fresh tumor tissues were transported in Medium 154CF (Gibco™, M-154CF-500) on ice to preserve viability. The tumor tissues were then minced with scalpels to pieces <1 mm$^3$ and transferred in 10 ml Medium 154CF with collagenase (5 mg/ml, (Gibco™, 17018029)) to digest the tissue. The mixture was incubated for 90 min at 37 °C with manual shaking every 10 min. Then, 5 ml 0.05% trypsin-EDTA (Gibco™, 25300-054) was added to the mixture and the digestion cocktail was incubated again for 10 min at 37 °C. After incubation, the mixture was centrifuged at $500 \times g$ for 5 min at 4 °C. In all, 5 ml 10% fetal bovine serum (FBS, (VWR, S181B-500)) was added to the pellet to inhibit the digestion enzymes. The mixture was centrifuged at $500 \times g$ for 5 min at 4 °C. Cells were suspended in 10 ml MACS buffer (2% FBS, 2 mM EDTA (Promega, V4231) diluted in phosphate-buffered saline (PBS) (Bichsel, 100 0 324)). The cell suspension was then filtered through a 70-μm cell strainer (Plexus-Santé, PL000223). Afterward, the cells were stained with SYTO™ 13 Green Fluorescent Nucleic Acid Stain (Invitrogen, S7575) and SYTOX™ Orange Nucleic Acid Stain (Invitrogen, S11368) for viability assessment. Stained cells were suspended in MACS buffer. Single living cells (SYTO™ green$^{Pos}$, SYTOX™ orange$^{Neg}$) were then FACS sorted and collected in a collecting medium (10% FBS, 2 mM EDTA in PBS). Sorted cells were centrifuged at $500 \times g$ for 5 min at 4 °C. The cells were then suspended in 10% FBS and counted. The cell suspension was centrifuged at $500 \times g$ for 5 min at 4 °C. Cells were suspended in the appropriate volume of 10% FBS to obtain a final concentration of approximately 1000 cells/μl.

### scRNA-seq processing (emulsion/library preparation/sequencing)

Single-cell mRNA capture and sequencing were performed immediately after reaching the optimal cell suspension by the Lausanne Genomic Technologies Facility (GTF, https://wp.unil.ch/gtf/) using the Chromium Next GEM single cell 3′ v3.1 reagent kit (10× Genomics) following the manufacturer's protocol. The target number of captured cells was 10,000 per sample. Sequencing libraries were prepared per the manufacturer's protocol. Paired-end sequencing was performed on Illumina HiSeq 4000 (Hiseq Control Software, v.3.4.0) (for BCC1 and BCC2) and NovaSeq 10× (Novaseq Control Software v.1.7.5) (for BCC3, BCC4 and BCC5) devices using HiSeq 3000/4000 SBS Kit reagents according to 10x Genomics recommendations (28 cycles read1, 8 cycles i7 index read, and 91 cycles read2) at a median depth of 23,272 reads/cell for BCC1, 34,848 reads/cell for BCC2, 32,995 reads/cell for BCC3, 25,693 reads/cell for BCC4 and 29,406 reads per cell for BCC5.

Sequencing data were demultiplexed using the bcl2fastq2 Conversion Software (v. 2.20, Illumina). Raw sequencing data provided as fastq files were processed using the count function of the Cell Ranger pipeline (v.5.0.1, 10× Genomics (https://support.10xgenomics.com/single-cell-gene-expression/software/downloads/5.0)). The count function allowed us to demultiplex sequencing reads to individual cells, to align the reads to the human GRCh38 genome reference (refdata-gex-GRCh38-2020-A), and to generate filtered matrices of gene counts by cell barcodes.

### scRNA-seq data analysis

Filtered gene by cell barcode counts matrices were imported into R (v. 4.1.0, https://www.R-project.org/) for further analyses using the Seurat[66] package (v. 4.0.4). We first filtered the cells to only retain cells that had between 500 and 6000 detected genes, <20% of reads mapping to mitochondrial genes, and less than 20% of reads mapping to dissociation protocol-related genes[67]. The single cells of each sample were first processed separately: Raw counts were normalized using the NormalizeData function using the "LogNormalize" method and a scale factor of 10,000. The top 2000 variable genes were selected using the FindVariableFeatures function with the "vst" selection method. For visualization purposes, the five samples were integrated by using the FindIntegrationAnchors and IntegrateData functions with default parameters. Dimensionality reduction was performed by first scaling and centering the integrated data, running a principle component analysis (PCA), followed by a Uniform Manifold Approximation and Projection (UMAP) dimensional reduction on 25 principal components. Finally, cells were clustered using the sheared nearest neighbor (SNN) modularity optimization-based clustering algorithm implemented in the FindNeighbors and FindClusters functions, with 25 principal components and a resolution parameter of 0.8. The expression level of canonical marker genes[21–23] was used to identify the biological cell types present in each cluster of the total population.

Cell clusters expressing unexpected marker combinations (e.g., *KRT14* and *CD3E*) were considered doublets and manually removed. To assess the degree of sample mixing performed by the dataset integration, the Local Inverse Simpson's Index (LISI) score was calculated using the published code[68] and the lisi (v.1.0) package.

When subclustering, the same analysis method as described above was applied, including dimensionality reduction on 25 principal components and clustering, with resolution parameters of 1.0 and 0.3 for tumor cells and fibroblasts, respectively. Subclusters with an abnormal distribution of detected genes or abnormal distribution of mitochondrial/ribosomal gene percentages were excluded for further analysis.

We used Wilcoxon rank sum tests to determine which genes were differentially expressed between two groups, as implemented in the FindMarkers function of the Seurat package, with default parameters. Differentially expressed genes were determined between the defined High (TC12, TC13, TC14, TC15) and Low (TC1, TC2, TC3, TC4) infiltrative groups of the tumor cell subpopulations, as well as between the defined High (FC4) and Low (FC1) infiltrative clusters of the fibroblast subpopulations.

Over-representation analysis of Gene Ontology biological process gene sets[69] was performed using the gseGO function implemented in the clusterProfiler (v.4.0.5) and the org.Hs.eg.db (v.3.13.0) packages.

Using the AddModuleScore function implemented in the Seurat package, we calculated the average expression per cell of genes belonging to several gene signatures (Supplementary Data 7). This function calculates the average expression of the genes in a signature, subtracted by the average expression of control genes that are selected at random among genes with similar expression levels as the genes included in the signature of interest[28].

Using the cor.test function implemented in the stats package (v.4.1.0), we calculated the correlation coefficient $R$ and the $p$ value

between $T^{INF}$ and $T^{NOD}$ signatures in the tumor cell subpopulations and between the $S^{INF}$ and $S^{NOD}$ signatures in the fibroblast subpopulations.

## Copy number variation analysis

To infer genomic copy number structure, we performed InferCNV (v.1.8.1) as per the developer's suggestions using standard parameters (https://github.com/broadinstitute/inferCNV). We used a cutoff of 0.1 for the minimum average read counts per gene among reference cells. We used T cells as an internal reference control.

## Trajectory and pseudotime analysis

Pseudotime calculations were performed using the R package Monocle 3 (v.1.0.0)[35]. We first imported the tumor cells Seurat object and adapted it to obtain a Monocle3 object using the new_cell_data_set function. Then, we calculated 100 principal components using the preprocess_cds function of Monocle 3 package. Next, a UMAP was generated with the reduce_dimension (default parameters), and cells were clustered using the cluster_cells function, with a resolution parameter of 0.00025. Then UMAP embeddings and clustering at resolution 0.8 previously calculated using Seurat were transferred to the Monocle3 object. Tumor cells were subjected to trajectory analysis using the learn_graph function with default parameters except for use_partition = FALSE. A pseudotime value was assigned to each cell using the order_cells function, selecting TC1 as the root state. Finally, genes differentially expressed along the tumor cell trajectory were identified using Moran's *I* test implemented in the graph_test function, using the principal graph as the selected neighbor graph[35]. Branch-point variations in expression are not indicated in graphs showing gene expression along the pseudotemporal trajectory.

## GeoMx digital spatial profiling

Six nodular and 6 infiltrative BCC samples were selected and evaluated using the GeoMx Cancer Transcriptome Atlas with 1812 RNA targets (https://www.nanostring.com/products/geomx-digital-spatial-profiler/geomx-rna-assays/geomx-cancer-transcriptome-atlas/). Spatial transcriptomics analysis included up to 24 regions of interest (ROI) per tumor type, divided into tumor and stroma areas of illumination (AOI).

For slide preparation, we followed the GeoMx DSP slide preparation user manual (MAN-10087-04). In brief, FFPE tissue sections of 5-μm were mounted on positively charged histology slides by grouping 3 patients per slide. Sections were incubated at 65 °C for 1 h. Slides were deparaffinized in 3 xylene baths of 5 min, then rehydrated in ethanol gradient: 2 baths of 5 min in 100% EtOH, followed by 5 min in 95% EtOH. Slides were then washed with PBS 1×. Antigen retrieval was performed in Tris-EDTA pH 9.0 at 100 °C for 15 min at low pressure. Slides were first dived into hot water for 10 s, and then into Tris-EDTA buffer. The cooker vent stayed open during the procedure to ensure low pressure. Slides were then washed in PBS 1×, incubated in proteinase K in PBS (1 μg/ml) for 15 min at 37 °C, and washed again with PBS 1×. Tissues were post-fixed in 10% neutral-buffered formalin (NBF) for 5 min, washed twice for 5 min in NBF stop buffer (0.1 M Tris Base, 0.1 M Glycine), and finally once in PBS 1×. The RNA probe mix (CTA (https://www.nanostring.com/products/geomx-digital-spatial-profiler/geomx-rna-assays/geomx-cancer-transcriptome-atlas/), a pool of in situ hybridization probes with UV photocleavable oligonucleotide barcodes) was placed on each section and covered with HybriSlip Hybridization Covers. Slides were then incubated for hybridization overnight at 37 °C in a Hyb EZ II hybridization oven (Advanced Cell Diagnostics). The day after, HybriSlip covers were gently removed and 25-min stringent washes were performed twice in 50% formamide and 2× Saline Sodium Citrate (SSC) at 37 °C. Tissues were washed for 5 min in 2× SSC, then blocked in Buffer W (Nanostring Technologies) for 30 min at room temperature in a humidity chamber. Next, tissues were stained with panCK-532 (clone AE1 + AE3) (Novus, NBP2-33200) at 1:20

and SYTO 13 at 1:10 (Thermo Scientific S7575) in Buffer W for 1 h at room temperature. Slides were washed twice in fresh 2× SSC and then loaded on the GeoMx Digital Spatial Profiler (DSP).

For GeoMx DSP sample collections, entire slides were imaged at ×20 magnification and morphologic markers were used to select ROI using organic shapes. Automatic segmentation of ROI based on panCK$^{pos}$ markers was used to define AOIs, allowing to separate tumor cells (panCK$^{pos}$) from adjacent stromal cells (panCK$^{neg}$). A total of 95 AOIs were exposed to 385 nm light (UV), releasing the indexing oligonucleotides that were collected with a microcapillary and deposited into a 96-well plate for subsequent processing. The indexing oligonucleotides were dried overnight and resuspended in 10 μl of DEPC-treated water.

Sequencing libraries were generated by PCR from the photo-released indexing oligos and AOI-specific Illumina adapter sequences. Unique i5 and i7 sample indexes were added. Each PCR reaction used 4 μl of indexing oligonucleotides, 4 μl of indexing PCR primers, 2 μl of Nanostring 5× PCR Master Mix. Thermocycling conditions were 37 °C for 30 min, 50 °C for 10 min, 95 °C for 3 min; 18 cycles of 95 °C for 15 s, 65 °C for 1 min, 68 °C for 30 s; and 68 °C for 5 min. PCR reactions were pooled and purified twice using AMPure XP beads (Beckman Coulter, A63881), according to the manufacturer's protocol. Pooled libraries were paired-sequenced at 2 × 27 base pairs and with a unique dual indexes workflow on an Illumina HiSeq 4000 instrument (as previously described). HiSeq-derived FASTQ files for each sample were compiled for each compartment using the bcl2fastq program of Illumina, and then demultiplexed and converted to digital count conversion (DCC) files using the GeoMx DnD pipeline (v.1) of Nanostring according to manufacturer's pipeline. DCC files were imported back into the GeoMx DSP instrument for QC and data analyses using GeoMx DSP analysis suite version 2.2.0.111 (Nanostring). A minimum of 10,000 reads were required for each sample. Probes were checked for outlier status by implementing a global Grubb's outlier test with alpha set to 0.01. The counts for all remaining probes for a given target were then collapsed into a single metric by taking the geometric mean of probe counts. For each sample, an RNA-probe-pool-specific negative probe normalization factor was generated on the basis of the geometric mean of negative probes in each pool. To ensure good data quality, we calculated the 75th percentile of the gene counts (that is, the geometric mean across all non-outlier probes for a given gene) for each AOI, and normalized to the geometric mean of the 75th percentile across all AOIs to give the upper quartile (Q3) normalization factors for each AOI. The distribution of these Q3 normalization factors was then checked for outliers (Supplementary Fig. 2b). DEGs and PCA analyses were performed on the GeoMx DSP instrument software.

## Fluorescence in situ hybridization (FISH)

FFPE skin blocks were cut and 5-μm sections were mounted onto Superfrost Plus® microscope slides FFPE skin sections were heated 10 min at 60 °C. Slides were deparaffinized in 2 xylene baths of 3 min, then rehydrated in an ethanol gradient from 100% EtOH (2 baths of 3 min), followed by 95% EtOH (3 min) and 70% EtOH (3 min). Slides were then washed in distilled water. RNAscope® Hydrogen Peroxide (ACD, 322335) was added to the tissue sections and slides were incubated for 10 min at room temperature. Slides were then washed for 5 min in distilled water followed by 5 min in PBS 1×. Antigen retrieval was done in Tris-EDTA pH 9.0 at 100 °C for 10 min, followed by cooling for 30 min at room temperature. Slides were washed twice for 5 min in PBS 1×. Slides were then immersed for 20 s in 0.012% Triton X-100 and washed twice for 5 min in PBS 1×. Tissue sections were then surrounded with an ImmEdge Pen (Vector Laboratories, H-4000). Protease treatment, probe hybridization, amplifications, and stainings were performed with the RNAscope® Multiplex Fluorescent Reagent Kit v2 Assay kit (ACD, 323100) according to the manufacturer's instructions. The following probes were used: RNAscope® Probe-Hs-

FN1-C3 (310311-C3), RNAscope® Probe-Hs-COL1A1-C2 (401891-C2), RNAscope® Probe-Hs-KRT6A-C3 (520721-C3), RNAscope® Probe-Hs-MPPED1 (426131), RNAscope® Probe-Hs-INHBA-C2 (415111-C2), RNA-scope® Probe-Hs-INHBA (415111), RNAscope® Probe-Hs-TGFB1 (400881), RNAscope® Probe Hs-CLDN4-C2 (421041-C2), RNAscope® Probe-Hs-PVRL1 (403071), RNAscope® Probe-Hs-DSG3 (470181), RNA-scope® Probe-Hs-POSTN (409181). Opal 570 Reagent (AKOYA Bioscience, OP-001003), and Opal 650 Reagent (AKOYA Bioscience, OP-001005) were used for fluorescence visualization. The slides were acquired with Pannoramic 250 slide scanner and processed using the CaseViewer 2.4 and (Fiji Is Just) ImageJ softwares.

## Immunofluorescence and FISH co-detection

FFPE skin blocks were cut and 5-μm sections were mounted onto Superfrost Plus® microscope slides FFPE skin sections were heated 10 min at 60 °C. Slides were deparaffinized in 2 xylene baths of 3 min, then rehydrated in an ethanol gradient from 100% EtOH (2 baths of 3 min), followed by 95% EtOH (3 min) and 70% EtOH (3 min). Slides were then washed in distilled water. RNAscope® Hydrogen Peroxide (ACD, 322335) was added to the tissue sections and slides were incubated for 10 min at room temperature. Slides were then washed for 5 min in distilled water followed by 5 min in PBS 1×. Antigen retrieval was done in 1× Co-detection Target Retrieval buffer (ACD, 323163) at 100 °C for 10 min, followed by cooling for 30 min at room temperature. Slides were then washed twice 1 min in distilled water and 2 min in 1× PBS with 0.1% Tween-20 (PBS-T) (PBS: Bichsel, 100 0 324, Tween-20: Sigma, P1379-500ML). Each tissue section was then covered with the primary antibody solution (Monoclonal antibody to human cytokeratin (pan) (clone Lu-5) (BMA Biomedicals, T-1302) diluted 1:250 in Co-Detection Antibody Diluent (ACD, 323160)) and incubated overnight at 4 °C in a humidified box. Following primary antibody incubation, slides were washed twice in PBS-T for 2 min. Slides were then fixed for 30 min in Buffered Zinc Formalin (Thermo Scientific, 5701ZF) at room temperature. After fixation, slides were washed four times in PBS-T for 2 min. Protease treatment, probe hybridization, amplifications, and stainings were performed with the RNAscope® Multiplex Fluorescent Reagent Kit v2 Assay kit (ACD, 323100) according to the manufacturer's instructions. After the final in situ hybridization horseradish peroxidase (HRP)-blocker step, each tissue section was covered with the secondary antibody dilution (goat F(ab')2 fragment IgG (H + L) anti-mouse Alexa Fluor 488 antibody (Life Technologies, A11017) diluted 1:500 in Co-Detection Antibody Diluent) for 30 min at room temperature. Slides were then washed twice for 2 min in PBS-T. Finally, slides were mounted with a mounting medium with DAPI-Aqueous Fluoroshield (abcam, ab104139). The slides were acquired with Zeiss LSM 700 confocal microscope and processed using the ZEN 2.3 lite and (Fiji Is Just) ImageJ softwares.

## Statistical analyses

Data represent results from three or more independent biological samples unless otherwise specified. Bar and line graphs results reflect the mean with standard deviation (SD). For spatial sequencing, statistics were conducted between infiltrative and nodular Tumor, respectively Stroma AOI. Spatial sequencing samples in the same disease category can be considered biological replicates. Statistical comparisons were performed using unpaired two-sided Student's $t$ test or Mann–Whitney $U$-test, according to the variances. Correlation analyses were calculated by Spearman correlation test. The softwares used for statistical analyses are GraphPad Prism (version 8.3.0) and GeoMx DSP analysis suite (version 2.2.0.111, Nanostring). $p$ Values are mentioned in the figures. Normal distribution was observed for all data. No technical replication was performed. No statistical method was used to predetermine sample size. No samples were excluded from the analyses. The experiments were not randomized.

## Reporting summary

Further information on research design is available in the Nature Research Reporting Summary linked to this article.

## Data availability

The scRNA-seq data generated in this study have been deposited in the Gene Expression Omnibus database under accession code GSE181907. The DSP data generated in this study have been deposited in the Gene Expression Omnibus database under accession code GSE210648. The following databases and datasets were used in this study: GRCh38 human reference genome reference (refdata-gex-GRCh38-2020-A) (https://support.10xgenomics.com/single-cell-gene-expression/software/downloads/latest), GeoMx® Cancer Transcriptome Atlas (https://www.nanostring.com/products/geomx-digital-spatial-profiler/geomx-rna-assays/geomx-cancer-transcriptome-atlas/). The remaining data are available within the Supplementary Information or Source data file. Source data are provided with this paper.

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

## Acknowledgements

This work was supported by an SNSF fellowship PZOOP3-185926 (F.K.), a UNIL-CHUV fellowship (F.K.), and the PROMEDICA Stiftung (F.K.). We thank the Genomic Technologies Facility (GTF) and Immune Landscape Monitoring Onco-CHUV for the scRNA-seq and GeoMx DSP library preparation and sequencing. We thank the Swiss Institute of Bioinformatics (SIB) for help with bioinformatics analyses. S.W. and M.G. are members of the SKINTEGRITY.CH collaborative research consortium.

## Author contributions

L.Y. and F.K. conceived the study, performed and analyzed the experiments, and wrote the manuscript with comments from co-authors. L.Y. and T.W. performed bioinformatics analyses. S.T.-R. performed the digital spatial profiling using GeoMx. J.D.D. and M.G. provided access to the tissue biobank. C.P.-B., M.C., and S.W. contributed to discussions throughout the study.

## Competing interests

The authors declare no competing interests.
