## [Peer Review File · Nature Communications]

Integrated multi-omics reveals cellular and molecular interactions governing the invasive niche of basal cell carcinomaReviewers' Comments:

Reviewer #1:

Remarks to the Author:

The manuscript takes an integrative approach, combining single-cell RNAseq of two human invasive BCCs and spatial transcriptomics (GeoMx) of six invasive and six nodular human basal cell carcinomas to investigate the genes and cells driving the invasive front of BCC. The manuscript is technically sound, follows standard approaches to analysis of scRNAseq, is highly informative and original. They take a slightly different "reverse" approach from others, using transcriptomic signatures generated from GeoMx-defined spatially-restricted areas of interest mapped onto scRNA-seq data. This analysis shows that the tumor invasive front is characterized by a collective migration phenotype that interacts intimately with activated CAFs that secrete remodeling ECM proteins and cytokines/chemokines. The investigators find that Activin A, is synthesized and secreted from the tumor cells of the invasive front and that adjacent CAF subpopulations are enriched for activin-A responsive transcripts. The manuscript questions earlier assumptions that EMT signatures seen in bulk RNAseq relate to EMT of tumor cells, whereas their integrated scRNAseq and GeoMx suggests that the EMT signature is based in activated CAFs. They also re-assess their own and others' studies suggesting involvement of TGFB1 in invasion of BCCs. Some of the findings postulated from these molecular approaches are confirmed by FISH analysis of specific genes on the various human BCCs samples. Overall, the manuscript is of high quality and relatively clearly written.

I have a few general comments, requests, and questions to make the manuscript clearer to the reader.

First, a question on the origin of the CAFs. Does the scRNAseq data provide insight into whether any of the CAFs may arise from tumor cells i.e. do any cells harbor tumor-specific mutations?

With respect to the findings on activin A, which activin receptors are expressed in CAFs that might instigate the effects of this cytokine? The investigators mention that TGF β inhibitors have been relatively unsuccessful in the clinical, but many of these inhibit the activin type I receptor, ACVRL1B/Alk4. Any comments?

Since the investigators conclude that Activin A and TGF β 1 show different pseudo-temporal trajectories, can they show different spatial expression patterns by FISH?

Although the investigators cite Ji et al 2020 Cell (on SCC) they do not mention the strikingly similar finding, albeit in another skin tumor type, SCC, that INHBA (activin A) is also found as a "Tumor-specific keratinocyte" population. I feel that the significance of these similar findings should be included in the Discussion.

Specific points for enhancing clarity:

Line 189. I think that "7 clusters" should read "8 clusters".

Sentence Line 190-1 is not clear. Should (TC3, TC5, TC6) be inserted after "cycling? And TC4 and TC7 after "small tumor".

Where monochrome gene expression intensity is utilized to show differential expression of genes across manifolds, it may be better to show bi-chrome gradients to emphasize differential expression. e.g, Figure 4, b,d,f and g.

Some of the statements in the Results do not appear to be supported by the data e.g. line 199-205, based on Figure 4c, it does not appear that the TC0/TC1 clusters show higher expression of KTR5 or KRT17 than TC02, and data for KRT1 and FLG are not shown.

Figure 3a Cluster numbers in key would benefit from the prefix C, likewise for Figure 4A prefix TC, and for Fig 5a prefix FC.

Line 298: replace "coding for the INHBA subunit of other TGFB family members" with "coding for the

inhibin (beta)A subunit of other TGFB super-family members".
Line 306 add "homodimeric" before "Activin A"

Finally, in Figure 1 I suggest using different nomenclature for the tumor samples in Fig 1a, such as BCC – followed by ID number, since "LyXXXX" looks like a drug name (Eli Lilly).

Reviewer #2:

Remarks to the Author:

The manuscript of Yerly et al. presents a combined analysis of scRNA sequencing of 2 infiltrative human BCCs and spatial transcriptomics of 12 BCCs. They report the enrichment of CAF signature in infiltrative compared to nodular BCC, which might explain an increased EMT signature in infiltrative BCC. Moreover, they observed a spatial organization of BCC with transcriptional reprogramming of cancer cells at the invasive front of the tumor.

Major comments:

1. The comparison of tumor sites at tumor-stroma interaction area between nodular and infiltrative BCC was performed. Was that difference representing the difference between nodular and infiltrative histotypes in general or it was specific tumor-stromal interaction areas?
2. The removal of non-specific genes from tumor DEGs was performed. Why these genes appeared among tumor DEGs? Is it because some tumor AOIs (particularly infiltrative BCC) were not pure and were contaminated by the stromal cells? For example, it might be useful to do PCA with all AOIs with all DEGs.
3. It is not clear why tumor or stromal DEGs were compared between tumor and stromal cells (Fig 1d,e).
4. On the figure 3a tumor cells do not form a separate cloud from keratinocytes which is unexpected.
5. Why typical markers of BCC such as: PTCH1, GLI1, GLI2, HHIP, MYCN, etc. were not used for the separation of BCC tumor cells from normal keratinocytes. Bulk RNAseq allows to separate BCC from normal keratinocytes with certainty. Instead, a signature composed of genes, which are differentially expressed between nodular and infiltrative BCC, were used as a signature of tumor cells.
6. Pseudotime analysis is not explained in detail and does not sound very convincing. Further validation of Activin A expression by tumor cells and of Activin A-induced signature enrichment in CAFs is required.

Reviewer #3:

Remarks to the Author:

Yerly et al present a molecular analysis of basal cell carcinoma to identify determinants of invasion at the tumor-normal border. They perform scRNA-seq on two BCCs followed by AOI-focused targeted RNA-profiling of 6 infiltrative vs 6 nodular BCCs. They identify genes differentially expressed in the infiltrative vs. non-infiltrative tumor border and project DGE onto scRNA-seq finding that the signature is expressed in the presumed tumor cluster, while non-tumor (stromal" regions enrich in CAFs. In a rather arbitrary approach, they exclude some tumor clusters and define the expression of the "infiltrative" signature of 12 genes and define low, intermediate and high scoring cells/remaining clusters, and the high-scoring cluster also scores strongly for a differentiation signature. Similarly, CAFs have variable expression of an infiltration n signature derived from spatial profiling. They claim that, in a pseudotime analysis, there is progressive acquisition of an infiltrative signature of cancer cells. In a biased interaction analysis between presumed cancer cells and CAFs, they find increased expression of selected genes, such as INHBA and find this gene to be enriched in the infiltrative border, while INHBB is decreased. Yet, they conclude that biologically active activin secreted by CAFs mediates infiltrative behavior of tumor cells at the margin, and find a modest correlation to support this in their spatial data.

BCC is a common disease and understanding determinants of infiltrative behavior is of importance. Unfortunately, in the eyes of this reviewer, the study has multiple major flaws in the design, analysis and statistical evaluation, yet derives major biological conclusions.

1. Fundamentally, the study is utterly underpowered to support any claims made based on the single-cell data. All of the analyses are done on cells from two individuals, which even from simply looking at the provided H&Es are clearly different. Thus, multiple claims made throughout the paper might simply be explained by differences between these two individuals (specific examples given below). Even if these tumors were molecularly identical, there is simply no way of statistically evaluating any of the claims made by the authors throughout. While low power is pervasive in all single-cell studies to date, and the goal here is not to achieve power as this is practically and financially inconceivable, the authors have to make an effort to collect additional samples, and substantiate all of their analyses with the best possible statistical tests. One should not forget that many cells per patient do not replace the need for including more patients.

2. The study has major analytical issues. First, fundamentally, the authors have not proven that what they assume to be cancer cells are in fact cancer cells. To do so, they should use now well-established methods for inferring chromosomal aberrations (Tirosh Science 2016). Second, they claim that the samples are well-integrated. I disagree with this statement. The fact of the matter is that is that one patient (LY31120) contributes the vast majority of KRT14+ cells and this includes a major cluster that is almost completely made up from this patient. Additionally virtually all cells in the CAF cluster are from this one patient. The vast difference in cell numbers has a major impact on the integration (which is very stringent). Also, how do they quantify that the data is well integrated? They should apply a quantitative method (e.g. LISI score) to substantiate this, however, based on the imbalance among patients, and the small patient sample size, these scores will remain highly vulnerable. Third, virtually all analyses presented with the single-cell data have to be referenced against 1) patient of origin, 2) quality of cells (e.g. UMI or genes/cell count) to exclude trivial explanations for the variability they see among, e.g. low/intermediate/high-scoring genes for their infiltration signature. Fourth, they derive a spatial tumor signature, and divide this further into nodular vs invasive. They score the cancer cells for the whole signature and the stromal cells for the CAFs, and as expected, find a strong correlation in these analysis. However, they should also apply the spatialTINF signature to CAFs, and vice versa, stromal signatures to tumors. The reason being that these signatures may be redundantly expressed in CAFs and tumors cell and vice versa as previously shown (Tirosh Science 2016). Fifth, how do we know that these signatures are non-random? The authors should make an effort to apply proper statistical tests (in this case possibly a hypergeometric test) to determine the specificity of these for the sub-populations they define. Finally, while they exclude "small" cluster in the presumed tumor cell analyses, they do not apply the same standard to the analysis of CAFs, in which probably every sub-cluster would be considered "small" consisting of a few dozen cells each. We also know that most cells are coming from one patient, so again, there is a major bias of this entire analysis hinging on a small number of cells from a single individual.

3. Interpretation: given these flaws, I find it difficult to objectively assess whether any of the proposed biology is either novel or accurate.

4. Innovation: the authors propose that integrating spatial to single-cell data is novel. I disagree with this. First, they use an AOI based method measuring a limited set of genes, which limits their discovery potential. Second, there are dozens of studies published (or preprints) that present robust analytical frameworks for label transfer into each direction and also reference-free methods that enable inference of spatial and single-cell transcriptomics data and vice versa.

Minor:

The title, including "multi-omics" is rather strongly worded and makes claims about "skin cancer", which is a rather broad wording that is not exhaustively studied here.

Manuscript Number: NCOMMS-21-33101

“Integrated multi-omics reveals cellular and molecular interactions governing the invasive niche of skin cancer”

We thank the Reviewers for their contributive and encouraging remarks and have endeavored to address each point below.

Point-by-Point Rebuttal of Reviewer’s comments:

Reviewer #1, expertise in skin and TGF β signalling (Remarks to the Author): The manuscript takes an integrative approach, combining single-cell RNAseq of two human invasive BCCs and spatial transcriptomics (GeoMx) of six invasive and six nodular human basal cell carcinomas to investigate the genes and cells driving the invasive front of BCC. The manuscript is technically sound, follows standard approaches to analysis of scRNAseq, is highly informative and original. They take a slightly different “reverse” approach from others, using transcriptomic signatures generated from GeoMx-defined spatially-restricted areas of interest mapped onto scRNA-seq data. This analysis shows that the tumor invasive front is characterized by a collective migration phenotype that interacts intimately with activated CAFs that secrete remodeling ECM proteins and cytokines/chemokines. The investigators find that Activin A, is synthesized and secreted from the tumor cells of the invasive front and that adjacent CAF subpopulations are enriched for activin-A responsive transcripts. The manuscript questions earlier assumptions that EMT signatures seen in bulk RNAseq relate to EMT of tumor cells, whereas their integrated scRNAseq and GeoMx suggests that the EMT signature is based in activated CAFs. They also re-assess their own and others’ studies suggesting involvement of TGFB1 in invasion of BCCs. Some of the findings postulated from these molecular approaches are confirmed by FISH analysis of specific genes on the various human BCCs samples. Overall, the manuscript is of high quality and relatively clearly written.

I have a few general comments, requests, and questions to make the manuscript clearer to the reader.

First, a question on the origin of the CAFs. Does the scRNAseq data provide insight into whether any of the CAFs may arise from tumor cells i.e. do any cells harbor tumor-specific mutations?

REPLY: We thank Reviewer #1 for this interesting suggestion. Based on the scRNA-seq, *KRT14*^{pos} tumor cells and CAFs segregate very well on the UMAP plot (Fig. 1b), with no intermediate population. We also looked at copy number variation inferred from scRNA-seq data (Fig. 1d). Compared to tumor cells, fibroblasts harbor much lower CNVs. Altogether, our scRNA-seq data do not support CAFs originated from tumors cells, although further experiments (like in vivo lineage tracing experiments) would be required to fully address the question.

This is now mentioned in the Results section (page 14, lines 289-292)

With respect to the findings on activin A, which activin receptors are expressed in CAFs that might instigate the effects of this cytokine?

REPLY: Activin A signals through type I (mainly ALK4, but also ALK7) and type II (ACTRIIA, ACTRIIB) transmembrane serine/threonine kinase receptors^{1,2}. Type II receptor binds the activin ligands and, once complexed, recruits, associates with and phosphorylates type I receptor. The most abundant activin, Activin A, typically signals through ACTRII/ALK4 heterodimer. Indeed, skin fibroblasts and skin CAFs responsiveness to Activin A was previously shown to be mediated by ALK4³⁻⁵. Based on scRNA-seq data, we confirmed *ACVRIIA/B* and *ALK4 (ACVRIB)* expression in BCC CAFs.

This is now shown in Supplementary Fig. 8 and mentioned in the Results section (page 15, lines330-334)

The investigators mention that TGF β inhibitors have been relatively unsuccessful in the clinical, but many of these inhibit the activin type I receptor, ACVRL1B/Alk4. Any comments?

REPLY: Indeed, some TGF β RI kinase inhibitors also inhibit ALK4. However, TGF β inhibitors have been relatively unsuccessful in the clinic for multiple other reasons, independently of their blocking effect on ALK4⁶. Therefore, the poor results obtained with TGF β inhibitors in the clinics so far do not preclude the use of Activin A/activin receptor inhibitors. Moreover, currently tested Activin A/activin receptor inhibitors, including recombinant follistatin (FST), neutralizing Activin A antibodies and soluble activin receptors, selectively target Activin A without affecting TGF β R1. Thus, they may differ in term of efficacy and adverse effects compared to TGF β R1 kinase inhibitors. Indeed, they showed some efficacy with mild-to-moderate adverse effects in various pre-clinical⁷⁻¹³ and clinical^{7,8} (Regeneron Announces Encouraging Garetosmab Phase 2 Results in Patients with Ultra-Rare Debilitating Bone Disease (prnewswire.com)) studies.

Still, TGF β signaling inhibitors have suffered from the dual pro- and anti-tumoral properties of TGF β signaling. Indeed, TGF β inhibition has been associated with the formation of benign tumors⁹. Similarly, Activin A has been shown to have both tumor-suppressive¹⁰⁻¹² and pro-tumorigenic effects^{4,13-17}. Although preclinical data suggest that Activin A signaling mostly promotes tumorigenesis in the skin^{4,18,19}, TGF β inhibitors and Activin A inhibitors may share similar challenges for proper efficiency in the clinics.

This is now discussed in the Discussion section (page 19, lines423-431).

Since the investigators conclude that Activin A and TGF β 1 show different pseudo-temporal trajectories, can they show different spatial expression patterns by FISH?

REPLY: Indeed, *INHBA* expression is restricted to highly infiltrative tumor cells clusters, while *TGFB1* expression shows a more diffuse pattern based on scRNA-seq (Fig. 6 and Supplementary Fig. 7). As requested by Reviewer #1, we confirmed different spatial expression using FISH on infiltrative BCC.

This is now shown in Supplementary Fig. 7.

Although the investigators cite Ji et al 2020 Cell (on SCC) they do not mention the strikingly similar finding, albeit in another skin tumor type, SCC, that *INHBA* (activin A) is also found as a “Tumor-specific keratinocyte “population. I feel that the significance of these similar findings should be included in the Discussion.

REPLY: We thank Reviewer #1 for this very contributive remark. Activin A is indeed known to be expressed by both BCC and SCC¹⁸. Remarkably, *INHBA* appears as a top marker gene for TSK, a population found at the tumor leading edge in the proximity of CAFs in SCC²⁰. These data together with our data support a common role for Activin A in governing the invasive niche of epithelial skin cancers.

Intriguingly however, SCC-TSK harbor EMT features, questioning the connection between *INHBA* expression and migration mode. Since last submission, we analyzed additional infiltrative BCC scRNA-seq samples (see Reviewer #3’s comment) which consolidated the highly infiltrative tumor cells clusters and their specific *INHBA* expression. While we see global enrichment for differentiation and collective migration, a very few *INHBA*^{POS} cells show EMT (Fig. 4f) and TSK (see hereunder) signatures enrichment.

Globally, it is very suggestive of few, rare, partial EMT tumor cells governing collective migration during development and cancer progression^{21,22}. However, additional studies with improved gene and spatial resolution will help to establish the relevance of such rare events.

These important, yet under investigation, points are mentioned in the Discussion section (page 18, lines 395-406).

Specific points for enhancing clarity: Line 189. I think that “7 clusters” should read “8 clusters”.

REPLY: The number of clusters was corrected according to the new UMAP (see Fig. 4a).

Sentence Line 190-1 is not clear. Should (TC3, TC5, TC6) be inserted after “cycling? And TC4 and TC7 after “small tumor”.

REPLY: As suggested by Reviewer #3, we subclustered non-cycling tumor cells and filtered out low quality clusters (showing abnormal distribution of detected genes/cell).

Where monochrome gene expression intensity is utilized to show differential expression of genes across manifolds, it may be better to show bi-chrome gradients to emphasize differential expression. e.g, Figure 4, b,d,f and g.

REPLY: We thank Reviewer #1 for this contributive remark. We modified UMAP illustrating signature scores by changing from monochrome to bichrome gradients to improve the visibility.

This is now corrected in Fig. 3b-c, 4b, 4e-f,5b, 5e, 5g, 7a; Supplementary Fig. 2c-f, 5a, 5c, 5h, 6a, 6c.

Some of the statements in the Results do not appear to be supported by the data e.g. line 199-205, based on Figure 4c, it does not appear that the TC0/TC1 clusters show higher expression of KTR5 or KRT17 than TC02, and data for KRT1 and FLG are not shown.

REPLY: We thank Reviewer #1 for this contributive remark. In the revised version, we paid attention to carefully support our statements by showing the data.

This has been corrected accordingly in Fig. 4g-i.

Figure 3a Cluster numbers in key would benefit from the prefix C, likewise for Figure 4A prefix TC, and for Fig 5a prefix FC.

REPLY: We thank Reviewer #1 for this suggestion. We modified violin plots on Fig. 3b-c to show the expression by cell type. In the Supplementary Fig. 2, we changed the cluster names according to the suggestion of Reviewer #1.

Line 298: replace “coding for the INHBA subunit of other TGFB family members” with “coding for the inhibin (beta)A subunit of other TGFB super-family members”.

REPLY: This was corrected (page 14, line 306)

Line 306 add “homodimeric” before “Activin A”

REPLY: This was corrected (page 15, line 314)

Finally, in Figure 1 I suggest using different nomenclature for the tumor samples in Fig 1a, such as BCC – followed by ID number, since “LyXXXX” looks like a drug name (Eli Lilly).

REPLY: We thank Reviewer #1 for this suggestion. We changed the nomenclature of sample ID from LYXXXX to BCC1-5 (for the scRNA-seq samples) and DSP1-12 (for the GeoMX DSP samples).

This is now specified in Supplementary Fig. 1a.

Reviewer #2, expertise in skin cancer genomics (Remarks to the Author): The manuscript of Yerly et al. presents a combined analysis of scRNA sequencing of 2 infiltrative human BCCs and spatial transcriptomics of 12 BCCs. They report the enrichment of CAF signature in infiltrative compared to nodular BCC, which might explain an increased EMT signature in infiltrative BCC. Moreover, they observed a spatial organization of BCC with transcriptional reprogramming of cancer cells at the invasive front of the tumor.

Major comments:

1. The comparison of tumor sites at tumor-stroma interaction area between nodular and infiltrative BCC was performed. Was that difference representing the difference between nodular and infiltrative histotypes in general or it was specific tumor-stromal interaction areas?

REPLY: We thank Reviewer #2 for highlighting this important point.

To best identify the relevant tumor-stroma interactions at the invasive tip of BCCs on scRNA-seq, we sequenced well-defined nodular and infiltrative spatially-resolved tumor-stroma interface areas. To do so, ROIs were designed to specifically cover spatially-resolved tumor-stroma interface areas. Illustrative

ROIs (divided into tumor and stromal AOIs) are shown in Fig. 2a. Additional illustrations of the ROI selection process are available in:

https://zenodo.org/record/5849365?token=eyJhbGciOiJIUzUxMiIsImV4cCI6MTY0NDc5MzE5OSwiaWF0IjoxNjQyMTY1MzE5fQ.eyJkYXRhIjp7InJlY2lkIjo1ODQ5MzY1fSwiaWQiOjE5ODEwLCJybmQiOiI2YjIc4YzBjNiJ9.oGhZZIT6MXGaSmJ_a792MdGY8itmAlv88VNtEoX90tfUhrkKRoiKTuTT2GU6s4T75CaH6-wl1hJBOZx3yDmHVQ

In the way they were designed, the inferred signatures are thus meant to be specific for the histotype (nodular versus infiltrative) at the spatially-resolved interface area.

Reviewer #2 wonders to what extent these spatial signatures reflect specific tumor-stroma interactions or nodular/infiltrative general histotypes. As the morphology at the spatially-resolved interface area is the cardinal feature of the histotype classification of BCCs, we expect the interface signatures to partially reflect the general histotypes.

To further assess the interface specificity of our signatures, we should look at their specific enrichment in “interface” versus “non-interface” spatially-resolved areas of the tumors, which, in our opinion, goes beyond the scope of this study.

As much as possible, this was clarified in Fig. 2a, in the Introduction section (page 4, lines 65-66 and 74-75) and in the Results section (pages 7, lines 131-136).

2. The removal of non-specific genes from tumor DEGs was performed. Why these genes appeared among tumor DEGs? Is it because some tumor AOIs (particularly infiltrative BCC) were not pure and were contaminated by the stromal cells? For example, it might be useful to do PCA with all AOIs with all DEGs.

REPLY:

As suggested by Reviewer #2, we performed PCA with all AOIs for all DEGs (Supplementary Fig. 4d). In doing so, we observed significant segregation of T^{NOD} , T^{INF} , S^{NOD} and S^{INF} AOIs. Intriguingly, T^{INF} appeared closer to stroma AOIs compared to T^{NOD} , while S^{NOD} appeared closer to tumor AOIs compared to S^{INF} . This suggests that T^{INF} and S^{INF} as well as T^{NOD} and S^{NOD} transcriptomes may respectively cross-

contaminate each other. Indeed, when overlapping T^{NOD} and S^{NOD} DEGs as well as T^{INF} and S^{INF} DEGs, we observed 5 and 18 shared genes respectively.

This is now shown in Supplementary Fig. 4d-e.

To best avoid that cross-contamination would affect spatial signatures, we thus filtered the DEGs using scRNA-seq data from *KRT14*^{pos} clusters (for tumor AOIs) or *KRT14*^{neg} clusters (for stroma AOIs).

These important points, including the filtering approach, were clarified in the Results section (pages 8-9; lines 147-182).

3. It is not clear why tumor or stromal DEGs were compared between tumor and stromal cells (Fig 1d,e).

REPLY: Spatial Tumor and spatial Stroma DEG signatures were compared for enrichment in tumor and stroma AOI to assess their respective specificity.

This was clarified in the Results section (page 8-9; lines 155-167).

In addition (see Reviewer #3's comment), we assessed the specificity of individual Spatial T^{NOD}, T^{INF}, S^{NOD} and S^{INF} signatures for T^{NOD}, T^{INF}, S^{NOD} and S^{INF} AOIs respectively.

This is now mentioned in the Results section (page 9, lines 178-182) and shown in Supplementary Fig. 4g.

4. On the figure 3a tumor cells do not form a separate cloud from keratinocytes which is unexpected.

REPLY: We understand Reviewer #2's concern. However, normal keratinocytes and tumor cells indeed clustered separately, although with minimal overlap, in a very similar manner as previously shown by others²⁰. This was highlighted in Fig. 1b, where tumor cells and normal keratinocytes clusters are now shown in different colors.

5. Why typical markers of BCC such as: *PTCH1*, *GLI1*, *GLI2*, *HHIP*, *MYCN*, etc. were not used for the separation of BCC tumor cells from normal keratinocytes. Bulk RNAseq allows to separate BCC from normal keratinocytes with certainty. Instead, a signature composed of genes, which are differentially expressed between nodular and infiltrative BCC, were used as a signature of tumor cells.

REPLY: We thank Reviewer #2 for this suggestion. As requested, we added *PTCH1*, *GLI1*, *GLI2*, *HHIP*, and *MYCN* expression patterns, which all confirmed higher expression in the identified tumor cells clusters.

This is now shown in Fig. 1c and Supplementary Fig. 2b.

In addition, we inferred copy number variation (using the R package “infercnv”) from scRNA-seq data to further establish and consolidate the identification of the normal keratinocytes and tumor cells clusters.

This is now shown in Fig. 1d.

6. Pseudotime analysis is not explained in detail and does not sound very convincing.

REPLY: We understand Reviewer #2’s concern, given the relative scarce information we gave in the initial manuscript. Pseudotime analysis allows unsupervised ordering of the cells along a path according to their transcriptional profile²³. Importantly, pseudotime analysis was now performed on 5 scRNA-seq BCC samples. Remarkably, when starting from the TC1 Low cluster, the pseudotemporal trajectory progressed to Med Low, Med High and ended with High infiltrative clusters (Fig. 4g), recapitulating the inversed T^{NOD} and T^{INF} signatures enrichment seen in Fig. 4b-d and Supplementary Fig. 5a-d, and mirroring the histopathological progression seen in BCCs.

This is now explained in the Methods section (page 25; lines 544-558), in the Results section (pages 11-12, lines 234-247) and shown in Fig. 4g.

Importantly, we confirmed reduced basal/BCC features and increased differentiation/collective migration features along the pseudotime trajectory.

This is now shown in Fig. 4h-i.

Further validation of Activin A expression by tumor cells and of Activin A-induced signature enrichment in CAFs is required.

REPLY: We previously showed *INHBA* expression in highly infiltrative tumor cell clusters on scRNA-seq. By adding 3 additional BCC scRNA-seq samples, we further consolidated the restricted expression of *INHBA* in the highly-infiltrative tumor cell clusters.

This is now shown in Fig. 6b-c.

Using FISH, we confirmed the preferential expression of *INHBA* in the highly infiltrative areas compared to low infiltrative areas within BCCs. We also confirmed tumor cell origin of *INHBA* using panCK co-staining.

This is now shown in Fig. 6c-d.

Based on scRNA-seq data, we confirmed the expression of the Activin A receptor constituents, *ACTRIIA/B* and *ALK4 (ACVRIB)*, in BCC CAFs.

This is now shown in Supplementary Fig. 8.

We also confirmed the specific Activin A exposure signature enrichment in infiltrative CAFs using 3 additional BCC samples scRNA-seq.

This is now shown in Fig. 7a.

Using FISH, we confirmed the spatial correlation between tumor *INHBA* and adjacent stroma *FN1* or *POSTN* (two known Activin A target genes) expression in human BCCs.

This is shown in Fig. 7c-d and Supplementary Fig. 9a-b.

Based on qRT-PCR or bulk RNAseq, we and others previously identified *FN1*^{24,25}, *POSTN*²⁴, *CCN4*²⁴, *COL3A1*²⁴, *ADAMTS2*²⁴ and *LRRC15*²⁴ as major contributors of infiltrative BCCs. Remarkably, 4 out of these 6 genes with predominant stromal expression (Supplementary Fig. 6h) were previously identified as Activin A target genes^{3,4}.

This is now mentioned in the Results section (pages 16, lines 345-347).

Altogether these additional data consolidate Activin A signaling as a major contributor of the infiltrative niche.

Reviewer #3, expertise in single cell sequencing and spatial methods (Remarks to the Author):

Yerly et al present a molecular analysis of basal cell carcinoma to identify determinants of invasion at the tumor-normal border. They perform scRNA-seq on two BCCs followed by AOI-focused targeted RNA-profiling of 6 infiltrative vs 6 nodular BCCs. They identify genes differentially expressed in the infiltrative vs. non-infiltrative tumor border and project DGE onto scRNA-seq finding that the signature is expressed in the presumed tumor cluster, while non-tumor (stromal" regions enrich in CAFs. In a rather arbitrary approach, they exclude some tumor clusters and define the expression of the "infiltrative" signature of 12 genes and define low, intermediate and high scoring cells/remaining clusters, and the high-scoring cluster also scores strongly for a differentiation signature. Similarly, CAFs have variable expression of an infiltration n signature derived from spatial profiling. They claim that, in a pseudotime analysis, there is progressive acquisition of an infiltrative signature of cancer cells. In a biased interaction analysis between presumed cancer cells and CAFs, they find increased expression of selected genes, such as *INHBA* and find this gene to be enriched in the infiltrative border, while *INHBB* is decreased. Yet, they conclude that biologically active activin secreted by CAFs mediates infiltrative behavior of tumor cells at the margin, and find a modest correlation to support this in their spatial data.

BCC is a common disease and understanding determinants of infiltrative behavior is of importance. Unfortunately, in the eyes of this reviewer, the study has multiple major flaws in the design, analysis and statistical evaluation, yet derives major biological conclusions.

1. Fundamentally, the study is utterly underpowered to support any claims made based on the single-cell data. All of the analyses are done on cells from two individuals, which even from simply looking at the provided H&Es are clearly different. Thus, multiple claims made throughout the paper might simply be explained by differences between these two individuals (specific examples given below). Even if these tumors were molecularly identical, there is simply no way of statistically evaluating any of the claims made by the authors throughout. While low power is pervasive in all single-cell studies to date, and the goal here is not to achieve power as this is practically and financially inconceivable, the authors have to make an effort to collect additional samples, and substantiate all of their analyses with the best possible statistical tests. One should not forget that many cells per patient do not replace the need for including more patients.

REPLY: We understand Reviewer #3's concern on the relative low number of BCC samples processed for scRNA-seq. As suggested, we thus included 3 additional infiltrative BCCs processed for scRNA-seq (total 5 BCC samples). These 5 BCC scRNAseq samples were controlled for quality (number of counts, number of detected genes, percentage of mitochondrial gene expression, percentage of dissociation gene expression) and integration (LISI score). Importantly, we confirmed all major biological conclusions previously made.

Quality control and integration are shown in Supplementary Fig. 1c, and are mentioned in the Methods section (page 23, line 498-500) and the Results section (page 6, line 95-96).

2. The study has major analytical issues. First, fundamentally, the authors have not proven that what they assume to be cancer cells are in fact cancer cells. To do so, they should use now well-established methods for inferring chromosomal aberrations (Tirosh Science 2016).

REPLY: We thank Reviewer #3 for this contributive remark. As suggested, we inferred copy number variation from scRNA-seq data (using the R package "infercnv") to further establish and consolidate the identification of the normal keratinocytes and tumor cells clusters respectively.

This is now shown in Fig. 1d.

In addition, we confirmed preferential expression of various BCC tumor marker genes (*PTCH1*, *GLI1*, *GLI2*, *HHIP*, and *MYCN*) in the tumor cell cluster, as suggested by Reviewer #2.

This is now shown in Fig. 1c and Supplementary Fig. 2b.

Second, they claim that the samples are well-integrated. I disagree with this statement. The fact of the matter is that is that one patient (LY31120) contributes the vast majority of KRT14+ cells and this includes a major cluster that is almost completely made up from this patient.

REPLY: We understand Reviewer #3's concern on scRNA-seq integration. However, Reviewer #3 was probably misled by the superposition of the 2 samples in the previous Fig. 1b UMAP. Indeed, our 2 initial BCCs scRNA-seq integrated well as shown on the UMAP split by sample (see hereunder). The respective contribution of each BCC sample to each cluster is shown in bar graphs (see hereunder).

We additionally performed a correlation analysis between the cluster gene expressions of each sample using non-batch corrected gene expression values. This correlation analysis reinforces the existing similarity between the 2 initial samples (see hereunder). In particular, the tumor cell clusters 0 and 8 show a correlation coefficient of 0.96 (with a p-value $< 2.2 \cdot 10^{-16}$). The fibroblast cluster 9 show a correlation coefficient of 0.97 (with a p-value $< 2.2 \cdot 10^{-16}$).

Similarly, the 5 BCC scRNAseq samples showed satisfactory integration (see hereunder comments).

Additionally virtually all cells in the CAF cluster are from this one patient. The vast difference in cell numbers has a major impact on the integration (which is very stringent).

REPLY: Again, we understand Reviewer #3's concern on scRNA-seq integration. As mentioned above, Reviewer #3 was probably misled by the superposition of the samples in the previous Fig. 1b UMAP. Indeed, our 2 initial BCC scRNA-seq integrated well as shown hereunder (split by patient).

Also, how do they quantify that the data is well integrated? They should apply a quantitative method (e.g. LISI score) to substantiate this, however, based on the imbalance among patients, and the small patient sample size, these scores will remain highly vulnerable.

REPLY: As requested by Reviewer #3, to assess whether the cell types in our single-cell RNA-seq dataset were well-mixed across the 5 samples, we used the Local Inverse Simpson's Index (LISI) algorithm²⁶ generating a score. The LISI score distribution was similar among all cell types, except for B cells, meaning that cell types were globally well integrated.

This is now shown in Supplementary Fig. 3b-c.

We also illustrated the individual contribution of each patient to each cell type with a bar graph, further supporting satisfactory integration.

This is now shown in Fig. 1e.

Third, virtually all analyses presented with the single-cell data have to be referenced against 1) patient of origin, 2) quality of cells (e.g. UMI or genes/cell count) to exclude trivial explanations for the variability they see among, e.g. low/intermediate/high-scoring genes for their infiltration signature.

REPLY: In general, we started by filtering out low quality cells for each sample individually. In fact, we filtered out cells expressing more than 20 percent mitochondrial genes, cells expressing more than 20 percent dissociation-associated genes, and cells expressing less than 500 genes or more than 6000 genes.

The five samples filtered for high quality of cells were then integrated. After integration, the filtered samples demonstrated, as expected, homogenous number of detected genes per cells, number of counts per cells, percentage of mitochondrial genes and percentage of dissociation-associated genes.

This is now shown in Supplementary Fig 1c and mentioned in the Method section (page 23, lines 498-500).

Regarding patient of origin, we previously showed satisfactory integration of the 5 different samples.

This is now shown in Fig 1e and Supplementary Fig. 3b-c.

This is further illustrated hereunder by the UMAP plot containing all cell types / tumor cells / fibroblasts split by patient.

Fourth, they derive a spatial tumor signature, and divide this further into nodular vs invasive. They score the cancer cells for the whole signature and the stromal cells for the CAFs, and as expected, find a strong correlation in these analysis. However, they should also apply the spatialTINF signature to CAFs, and vice versa, stromal signatures to tumors. The reason being that these signatures may be redundantly expressed in CAFs and tumors cell and vice versa as previously shown (tirosh science 2016).

REPLY: We thank Reviewer #3 for this interesting comment. Indeed, adjacent tumor cells and CAFs may share transcriptional programs.

As suggested, we thus applied T^{NOD}/T^{INF} to CAFs and S^{NOD}/S^{INF} to tumor cells. As expected, global enrichment was higher in tumor cells than in CAFs for T^{NOD}/T^{INF} signatures. Reversibly, global enrichment was higher in CAFs than in tumor cells for S^{NOD}/S^{INF} signatures. Still, T^{NOD} and T^{INF} showed a discrete but inversed gradient when applied on CAFs. Similarly, S^{NOD} showed a discrete enrichment in the Low infiltrative tumor clusters (see hereunder). Altogether, it suggests that transcriptional programs are partially shared between adjacent tumor compartments, as previously shown²⁷.

Fifth, how do we know that these signatures are non-random? The authors should make an effort to apply proper statistical tests (in this case possibly a hypergeometric test) to determine the specificity of these for the sub-populations they define.

REPLY: We thank Reviewer #3 for this contributive remark. To address this question, we applied individually Spatial T^{NOD} , T^{INF} , S^{NOD} and S^{INF} signatures on T^{NOD} , T^{INF} , S^{NOD} and S^{INF} AOIs.

When looking into the 25% upper quartile for enrichment, we found significant specificity (calculated by hypergeometric test) for all four signatures and their respective compartment. Intriguingly, S^{NOD} signature showed significant specificity for T^{NOD} AOI as well, suggesting shared transcriptional programs, as suggested hereabove.

This is now shown in Supplementary Fig. 4g and mentioned in the Results section (page 9; lines 178-182).

Finally, while they exclude "small" cluster in the presumed tumor cell analyses, they do not apply the same standard to the analysis of CAFs, in which probably every sub-cluster would be considered "small" consisting of a few dozen cells each. We also know that most cells are coming from one patient, so again, there is a major bias of this entire analysis hinging on a small number of cells from a single individual.

REPLY: As previously mentioned, we included 3 additional scRNA-seq of infiltrative BCC and showed acceptable integration on the various clusters, including non-cycling tumor cells and fibroblasts. We now subclustered non-cycling tumor cells and fibroblasts similarly, by filtering out cell clusters with abnormal distribution of the number of detected genes per cell.

Subclustering is now shown in Fig. 4a and 5a and subclustering approach mentioned in the Methods section (page 23-24, lines 516-521)

3. Interpretation: given these flaws, I find it difficult to objectively assess whether any of the proposed biology is either novel or accurate.

REPLY: We are thankful to Reviewer #3 as the major biological conclusions were confirmed and consolidated by addressing the hereabove mentioned issues.

4. Innovation: the authors propose that integrating spatial to single-cell data is novel. I disagree with this. First, they use an AOI based method measuring a limited set of genes, which limits their discovery potential. Second, there are dozens of studies published (or preprints) that present robust analytical frameworks for label transfer into each direction and also reference-free methods that enable inference of spatial and single-cell transcriptomics data and vice versa.

REPLY: We actually do not claim to be the first to integrate spatial and single-cell data (reviewed in REF²⁸). Indeed, we referenced the few cardinal articles resolving scRNAseq and spatial transcriptomics data in cancer^{20,29,30}. Since first submission, Hunter MV et al published an additional study integrating spatial and single-cell data to decipher the architecture of the tumor microenvironment³¹, that we now cited. In contrast with these previous articles resolving scRNA-seq and spatial transcriptomics data in cancer, we used a slightly different approach, where, despite limited gene set (1812), a high number of spatially-restricted areas of interest in tumors are defined transcriptionally, integrated and then mapped onto scRNA-seq data to obtain cellular resolution.

This is now clarified/rephrased in the Abstract (page 2 lines 32-34) and Discussion section (page 17, lines 374-382; page 20, line 436).

Minor:

The title, including "multi-omics" is rather strongly worded and makes claims about "skin cancer", which is a rather broad wording that is not exhaustively studied here.

REPLY: In our opinion, "multi-omics" is justified by the dual sequencing approach (single-cell and spatial sequencing). We changed skin cancer for basal cell carcinoma, although, as suggested by previous publications (see Reviewer #1's comment), squamous cell carcinoma and basal cell carcinoma (by far the 2 predominant skin cancers) may share common INHBA-mediated interactions at the infiltrative niche.

This was modified in the Title (page 1, line 3)

In behalf of all authors,

François Kuonen

References

1. Tsuchida, K. *et al.* Activin signaling as an emerging target for therapeutic interventions. *Cell Commun. Signal.* **7**, 15 (2009).
2. Cangkrama, M., Wietecha, M. & Werner, S. Wound Repair, Scar Formation, and Cancer: Converging on Activin. *Trends Mol. Med.* **26**, 1107–1117 (2020).
3. Wietecha, M. S. *et al.* Activin-mediated alterations of the fibroblast transcriptome and matrix control the biomechanical properties of skin wounds. *Nat. Commun.* **11**, 2604 (2020).
4. Cangkrama, M. *et al.* A paracrine activin A–mDia2 axis promotes squamous carcinogenesis via fibroblast reprogramming. *EMBO Mol. Med.* **12**, e11466 (2020).
5. Fumagalli, M. *et al.* Imbalance between activin A and follistatin drives postburn hypertrophic scar formation in human skin. *Exp. Dermatol.* **16**, 600–610 (2007).
6. Teixeira, A. F., ten Dijke, P. & Zhu, H.-J. On-Target Anti-TGF- β Therapies Are Not Succeeding in Clinical Cancer Treatments: What Are Remaining Challenges? *Front. Cell Dev. Biol.* **0**, (2020).
7. Mendell, J. R. *et al.* A Phase 1/2a Follistatin Gene Therapy Trial for Becker Muscular Dystrophy. *Mol. Ther.* **23**, 192–201 (2015).
8. Raftopoulos, H. *et al.* Sotatercept (ACE-011) for the treatment of chemotherapy-induced anemia in patients with metastatic breast cancer or advanced or metastatic solid tumors treated with platinum-based chemotherapeutic regimens: results from two phase 2 studies. *Support. Care Cancer* **24**, 1517–1525 (2016).
9. Colak, S. & ten Dijke, P. Targeting TGF- β Signaling in Cancer. *Trends Cancer* **3**, 56–71 (2017).
10. Risbridger, G. P., Schmitt, J. F. & Robertson, D. M. Activins and Inhibins in Endocrine and Other Tumors. *Endocr. Rev.* **22**, 836–858 (2001).
11. Burdette, J. E., Jeruss, J. S., Kurley, S. J., Lee, E. J. & Woodruff, T. K. Activin A Mediates Growth Inhibition and Cell Cycle Arrest through Smads in Human Breast Cancer Cells. *Cancer Res.* **65**, 7968–7975 (2005).
12. Zhao, Y. *et al.* Oncogene-Induced Senescence Limits the Progression of Pancreatic Neoplasia through Production of Activin A. *Cancer Res.* **80**, 3359–3371 (2020).
13. Reader, K. L. & Gold, E. Activins and activin antagonists in the human ovary and ovarian cancer. *Mol. Cell. Endocrinol.* **415**, 126–132 (2015).

14. Lonardo, E. *et al.* Nodal/Activin Signaling Drives Self-Renewal and Tumorigenicity of Pancreatic Cancer Stem Cells and Provides a Target for Combined Drug Therapy. *Cell Stem Cell* **9**, 433–446 (2011).
15. Chen, L. *et al.* A NF- κ B-Activin A signaling axis enhances prostate cancer metastasis. *Oncogene* **39**, 1634–1651 (2020).
16. Rautela, J. *et al.* Therapeutic blockade of activin-A improves NK cell function and antitumor immunity. *Sci. Signal.* (2019) doi:10.1126/scisignal.aat7527.
17. Xie, D. *et al.* The effects of activin A on the migration of human breast cancer cells and neutrophils and their migratory interaction. *Exp. Cell Res.* **357**, 107–115 (2017).
18. Antsiferova, M. *et al.* Activin enhances skin tumourigenesis and malignant progression by inducing a pro-tumourigenic immune cell response. *Nat. Commun.* **2**, 576 (2011).
19. Schaper, I. D. *et al.* Development of Skin Tumors in Mice Transgenic for Early Genes of Human Papillomavirus Type 8. *Cancer Res.* **65**, 1394–1400 (2005).
20. Ji, A. L. *et al.* Multimodal Analysis of Composition and Spatial Architecture in Human Squamous Cell Carcinoma. *Cell* **182**, 497-514.e22 (2020).
21. Nieto, M. A., Huang, R. Y.-J., Jackson, R. A. & Thiery, J. P. EMT: 2016. *Cell* **166**, 21–45 (2016).
22. Jolly, M. K. *et al.* Implications of the Hybrid Epithelial/Mesenchymal Phenotype in Metastasis. *Front. Oncol.* **0**, (2015).
23. Cao, J. *et al.* The single-cell transcriptional landscape of mammalian organogenesis. *Nature* **566**, 496–502 (2019).
24. Villani, R. *et al.* Subtype-Specific Analyses Reveal Infiltrative Basal Cell Carcinomas Are Highly Interactive with their Environment. *J. Invest. Dermatol.* (2021) doi:10.1016/j.jid.2021.02.760.
25. Kuonen, F. *et al.* TGF β , Fibronectin and Integrin α 5 β 1 Promote Invasion in Basal Cell Carcinoma. *J. Invest. Dermatol.* **138**, 2432–2442 (2018).
26. Korsunsky, I. *et al.* Fast, sensitive and accurate integration of single-cell data with Harmony. *Nat. Methods* **16**, 1289–1296 (2019).
27. Tirosh, I. *et al.* Dissecting the multicellular ecosystem of metastatic melanoma by single-cell RNA-seq. *Science* **352**, 189–196 (2016).

28. Longo, S. K., Guo, M. G., Ji, A. L. & Khavari, P. A. Integrating single-cell and spatial transcriptomics to elucidate intercellular tissue dynamics. *Nat. Rev. Genet.* 1–18 (2021) doi:10.1038/s41576-021-00370-8.
29. Moncada, R. *et al.* Integrating microarray-based spatial transcriptomics and single-cell RNA-seq reveals tissue architecture in pancreatic ductal adenocarcinomas. *Nat. Biotechnol.* **38**, 333–342 (2020).
30. Thrane, K., Eriksson, H., Maaskola, J., Hansson, J. & Lundeberg, J. Spatially Resolved Transcriptomics Enables Dissection of Genetic Heterogeneity in Stage III Cutaneous Malignant Melanoma. *Cancer Res.* **78**, 5970–5979 (2018).
31. Hunter, M. V., Moncada, R., Weiss, J. M., Yanai, I. & White, R. M. Spatially resolved transcriptomics reveals the architecture of the tumor-microenvironment interface. *Nat. Commun.* **12**, 6278 (2021).

Reviewers' Comments:

Reviewer #1:

Remarks to the Author:

The authors have answered most of my questions, but answers raise some new questions.

My review below:

The Introduction should be edited for English grammar.

Is the term "split retraction" a standard term? If not, please rephrase or define.

"Indeed, previous studies of infiltrative BCCs suggested the role of ECM, ECM receptor and ECM remodeling genes like FN113,21, POSTN13, CCN413, COL3A113, ADAMTS21373 , LRRC1513, ITGA521, ITGAV22 and ITGB62274 ". Does this refer to differential role of ECM between nodular and invasive BCC, and if so in what direction? Or are these merely, molecular markers of BCCs versus normal epidermis?

Authors state that "Importantly, BCCs harbor a spectrum of nodular and infiltrative features, typically observed within individual tumors." But go on to investigate 6 invasive versus 6 nodular BCCs. In this case, how do they classify invasive versus nodular BCC? Does invasive BCC simply those with ≥ 1 region of invasion, whereas nodular has no invasive regions?

Line 134 "...to restrict our analysis to the tumor stroma interaction site in BCC." Please state INVASIVE BCC.

Lines 240-250 raise further questions, for example staining of infiltrative (versus nodular) tumors for CLDN4, NECTIN1 and DSG3, should be undertaken to demonstrate that cohesive migration is truly active within the invasive epithelial fingers where INHBA is expressed.

p271 – delete "Remarkably," since this would be as expected.

Pp 343-353: Even though the investigators interrogated 48 tumor areas and 47 adjacent stroma areas, this still only reflects six tumors. Validation for the hypothesis that collective epithelial migration and INHBA expression in invasive tumor cells drives invasion through induction of activin A-responsive genes in CAFs, would require a larger panel of both invasive and nodular tumor-stromal regions screened for expression of CLDN4, NECTIN1, DSG3, and/or INHBA by FISH or IHC. Data would be more compelling if the predictions from scRNA and DSP were shown to be reproducible in independent samples.

In Figure 6 c and d, only an infiltrative BCC is shown. Can they add images of a nodular BCC with same stains? Are these similar to the low-infiltrative region of the tumor?

Figure 6c. It is notable that INHBA is detected in stroma cells as well as epithelial cells, especially in low -infiltrative region. What kind of cells are these?

I cannot comment on the biostatistical analysis of the RNAseq.

Reviewer #2:

Remarks to the Author:

Majority of the concerns of this reviewer were adequately addressed.

Reviewer #3:

Remarks to the Author:

Yerly et al revised the initial manuscript; major highlights to improve the manuscript include.

1. addition of 3 more patients
2. additional analyses.

The authors attempted to address concerns raised by this reviewer. However, in doing so, they supported some of the critiques initially raised rather than removing them.

First, they now perform analysis of inferred CNV and show a representative figure of such in Figure 1d. The simple truth visible in this figure is that there are subpopulations of cells with variable CNV pattern, but there is a substantial portion of cells that simply do not show any CNV, yet they are listed in the cancer section. The authors argue that they orthogonally show that presumed cancer cells are expressing tumor markers. The truth of the matter, seen in Figure 1c and Suppl Figure 2b is that only cells labeled as "non-cycling tumor cells" express PTCH1 and MYC. Furthermore, Fig. S2c shows that only those cells score strongly for Hedgehog signaling. This suggests that cells labeled as "cycling tumor cells" may not be tumor cells. Furthermore, expression of KRT14 is not sufficient to differentiate healthy from malignant cells. Lastly, the truth of the data is that the "cycling" cells do not express markers of cell cycle, as shown by the authors in Fig 1c (e.g. MKI67). The interpretation of their data cannot be continued until they confidently achieve a situation in which they can without a doubt define cancer cells as such. They should demonstrate the CNV plots for all, and not only once case. My recommendation is also that they use only T cells as a reference for CNV inference in the test cells (e.g. cancer cells).

Second, perhaps the authors misinterpreted my initial suggestion "Third, virtually all analyses presented with the single-cell data have to be referenced against 1) patient of origin, 2) quality of cells (e.g. UMI or genes/cell count) to exclude trivial explanations for the variability they see among, e.g. low/intermediate/high-scoring genes for their infiltration signature." They address this by stating that they performed upfront quality control by filtering based on mitochondrial reads and gene complexity, and that those samples integrate (using stringent CCA). That is not the point. Even though they filter upfront, which is required, there remains high variability in quality within and across each patient (e.g. gene complexity may range from 500 to 6000 genes detected per cell as shown by the authors). Thus, upfront filtering will not be sufficient to control for variability in quality in specific analyses, where clustering of non-integrated data may be driven by within and across patient cell complexity. A simple way to address this is to show that clustering in multiple analysis is not associated with complexity (e.g. by projecting the range of UMIs/gene counts).

Reviewer #1 (Remarks to the Author):

The authors have answered most of my questions, but answers raise some new questions.

My review below:

The Introduction should be edited for English grammar.

REPLY: The Introduction was edited for English grammar.

Is the term “split retraction” a standard term? If not, please rephrase or define.

REPLY: We rephrased using the term “peritumoral cleft”.

This was corrected in the Introduction section (page 4; lines 67 and 70-71).

“Indeed, previous studies of infiltrative BCCs suggested the role of ECM, ECM receptor and ECM remodeling genes like FN113,21, POSTN13, CCN413, COL3A113, ADAMTS21373 , LRRC1513, ITGA521, ITGAV22 and ITGB62274 “. Does this refer to differential role of ECM between nodular and invasive BCC, and if so in what direction? Or are these merely, molecular markers of BCCs versus normal epidermis?

REPLY: We thank Reviewer #1 for this comment. *FN1, POSTN, CCN4, COL3A1, ADAMTS2 , LRRC15, ITGA5, ITGAV* and *ITGB6* were all shown to be preferentially expressed in infiltrative compared to nodular BCCs, using comparative bulk RNA-seq or IF studies¹⁻³.

We clarified this important point in the Results section (page 11; lines 254-257).

Authors state that “Importantly, BCCs harbor a spectrum of nodular and infiltrative features, typically

observed within individual tumors.” But go on to investigate 6 invasive versus 6 nodular BCCs. In this case, how do they classify invasive versus nodular BCC? Does invasive BCC simply those with ≥ 1 region of invasion, whereas nodular has no invasive regions?

REPLY: We thank Reviewer#1 for raising this point. BCCs were classified as nodular or infiltrative by independent dermatopathologists, classically based on the predominant histological morphology of their respective tumor-stroma interface. However, since nodular and infiltrative morphologies found at the tumor-stroma interface are not categorical entities but rather represent a continuum reflected by intra-tumor heterogeneity, we focused our analysis on spatially-resolved areas with characteristic morphological features within individual tumors.

Characteristic Infiltrative tumor-stroma interface regions were identified within infiltrative (according to dermatopathologist’s report) BCC sections, and compared to characteristic Nodular tumor-stroma interface regions identified within nodular (according to dermatopathologist’s report) BCC sections.

Thereby, we designed Tumor and Stroma interface signatures that are specific for Nodular and Infiltrative morphologies.

This was clarified in the Introduction section (page 4, lines 64-73) and in the Results section (pages 6-7, lines 132-136; page 7, line 142; page 8, lines 184-185).

Line 134 “...to restrict our analysis to the tumor stroma interaction site in BCC.” Please state INVASIVE BCC.

REPLY: The paragraph was rephrased, according to the previous comment (page 6, lines 132-136).

Lines 240-250 raise further questions, for example staining of infiltrative (versus nodular) tumors for *CLDN4*, *NECTIN1* and *DSG3*, should be undertaken to demonstrate that cohesive migration is truly active within the invasive epithelial fingers where *INHBA* is expressed.

REPLY: As suggested, we performed RNA FISH co-stainings of collective migration markers (*CLDN4*, *NECTIN1* and *DSG3*) and *INHBA*. Consistently with the sequencing data, we observed co-expression of collective migration markers and *INHBA* expression. Importantly, *INHBA* and collective migration markers were predominantly found at the tumor-stroma interface of infiltrative (compared to nodular) BCCs.

This is now shown in Supplementary Fig. 13a-f.

p271 – delete “Remarkably,” since this would be as expected.

REPLY: This was corrected in the Results section (page 10, line 213; page 12, line 277).

Pp 343-353: Even though the investigators interrogated 48 tumor areas and 47 adjacent stroma areas, this still only reflects six tumors. Validation for the hypothesis that collective epithelial migration and *INHBA* expression in invasive tumor cells drives invasion through induction of activin A-responsive genes in CAFs, would require a larger panel of both invasive and nodular tumor-stromal regions screened for expression of *CLDN4*, *NECTIN1*, *DSG3*, and/or *INHBA* by FISH or IHC. Data would be more compelling if the predictions from scRNA and DSP were shown to be reproducible in independent samples.

REPLY: As suggested, we performed RNA FISH co-stainings of collective migration markers (*CLDN4*, *NECTIN1* and *DSG3*) and *INHBA* as well as activin A-responsive genes and *INHBA* on tumor-stromal regions from additional nodular ($n \geq 4$) and infiltrative (≥ 5) BCCs. In this independent cohort (Methods section, page 21, line 467), we validated:

- preferential expression of tumor *INHBA* in High infiltrative regions (compared to nodular), shown in Fig. 6c-d-e
- significant positive correlation of tumor *INHBA* with collective migration markers in tumor cells, shown in Supplementary Fig. 13a-f
- significant positive correlation of tumor *INHBA* with activin A-responsive genes in surrounding CAFs, shown in Fig. 7b-c and Supplementary Fig. 15b-c

In Figure 6 c and d, only an infiltrative BCC is shown. Can they add images of a nodular BCC with same stains? Are these similar to the low-infiltrative region of the tumor?

REPLY: As suggested, we stained nodular BCCs for *INHBA* (see hereabove). As expected, we found low expression of *INHBA* in tumor cells located at the tumor-stroma interface, similarly to Low-infiltrative tumor-stroma interface regions found within infiltrative BCCs.

This is now shown in Fig. 6c-d-e.

Figure 6c. It is notable that *INHBA* is detected in stroma cells as well as epithelial cells, especially in low-infiltrative region. What kind of cells are these?

REPLY: *INHBA* expression is indeed detectable in stromal cells, although not especially in Low-infiltrative regions (see Figure 6d). Consistently with previous reports^{4,5}, we found stromal *INHBA* mostly expressed by fibroblasts, as shown by the co-expression of *COL1A2* (RNA FISH) (**Point-by-point_Figure 1**) .

Point-by-point_Figure 1: Stromal INHBA is mostly expressed by fibroblasts.

I cannot comment on the biostatistical analysis of the RNAseq.

Reviewer #2 (Remarks to the Author):

Majority of the concerns of this reviewer were adequately addressed.

Reviewer #3 (Remarks to the Author):

Yerly et al revised the initial manuscript; major highlights to improve the manuscript include.

1. addition of 3 more patients
2. additional analyses.

COMMENT: Since last submission, in addition to previous quality controls, we identified cell clusters expressing unexpected marker combinations (e.g. *KRT14* and *CD3E*). Those were considered as doublets and manually removed.

This is now mentioned in the Methods section (page 23, lines 532-533).

The authors attempted to address concerns raised by this reviewer. However, in doing so, they supported some of the critiques initially raised rather than removing them.

First, they now perform analysis of inferred CNV and show a representative figure of such in Figure 1d. The simple truth visible in this figure is that there are subpopulations of cells with variable CNV pattern, but there is a substantial portion of cells that simply do not show any CNV, yet they are listed in the cancer section.

REPLY: We understand Reviewer #3's concern. However, the CNV pattern we previously showed integrated tumor cells from the 5 BCC samples, so that, in addition to the expected intra-tumor CNV variability, we visualized the inter-tumor CNV variability. The CNV analysis previously included cycling cells that were actually composed of both tumor cells and keratinocytes.

As suggested, we thus performed the inferred CNV analysis on each patient individually for non-cycling *KRT14*^{pos} epithelial cells, taking T cells as reference. We excluded cycling cells as they were not considered for downstream analysis (see hereunder discussion).

In all 5 BCCs, epithelial cluster C0 displayed higher CNV compared to epithelial clusters C1-C2-C3. In particular, when excluding gene expression changes observed in all five BCC samples (and thus potentially reflecting cell lineage-dependent transcription rather than genomic structural changes), inferred CNVs were exclusively found in the epithelial cluster C0, with a few being shared by virtually all cells of the cluster.

This is now shown in Fig.1f and Supplementary Fig. 2a-d.

Based on combined Seurat clustering and inferred CNV analysis, we thus identified epithelial cluster C0 cells as Tumor cells, while epithelial clusters C1, C2 and C3 represent keratinocytes. This was further supported by their respective enrichment for tumor and normal epidermis marker genes/signatures respectively.

This is now shown in Fig. 1g and Supplementary Fig. 3a-e.

Altogether, we can reasonably assume that we identified and distinguished tumor cells from normal keratinocytes with a high degree of confidence.

The authors argue that they orthogonally show that presumed cancer cells are expressing tumor markers. The truth of the matter, seen in Figure 1c and Suppl Figure 2b is that only cells labeled as "non-cycling tumor cells" express PTCH1 and MYC. Furthermore, Fig. S2c shows that only those cells score strongly for Hedgehog signaling. This suggests that cells labeled as "cycling tumor cells" may not be tumor cells.

REPLY: We thank Reviewer #3 for raising this point that escape our attention as we essentially focused on the non-cycling tumor cells for downstream analysis. Indeed, cycling cells, identified by *MKI67* positivity, are actually composed of both *KRT14*^{pos}*PTCH1*^{pos} and *KRT14*^{pos}*PTCH1*^{neg} subpopulations, suggesting mixed cycling tumor cells and cycling keratinocytes (**Point-by-point_Figure 2**).

Point-by-point_Figure 2: Cycling cells are composed of $KRT14^{pos}/PTCH1^{pos}$ and $KRT14^{pos}/PTCH1^{neg}$ cells.

We thus renamed the corresponding cluster as “Cycling cells” instead of “Cycling tumor cells”.

This is now corrected in Fig. 1b-c.

Importantly, cycling cells were not considered for downstream analysis.

Furthermore, expression of KRT14 is not sufficient to differentiate healthy from malignant cells.

REPLY: We perfectly agree with Reviewer #3 to say that *KRT14* is not sufficient to differentiate healthy from malignant cells. Instead, we used *PTCH1*, *GLI1*, *GLI2*, *MYCN* and *HHIP* expression, KEGG Hedgehog signaling pathway and KEGG Basal cell carcinoma signatures which all showed preferential enrichment in non-cycling tumor cells (cluster C0) compared to non-cycling keratinocytes (clusters C1-C2-C3). In contrast, cluster C3 showed specific enrichment for hair follicle signatures (Bulge/Isthmus), cluster C1 specific enrichment for basal epidermis signatures, and cluster C2 gradual enrichment for spinous/granular epidermis signatures, consistently with their normal keratinocyte nature.

This is now shown in Supplementary Fig. 3a-e.

Lastly, the truth of the data is that the "cycling" cells do not express markers of cell cycle, as shown by the authors in Fig 1c (e.g. MKI67).

REPLY: We politely disagree with Reviewer #3's concern, as "Cycling cells" are actually all *MKI67* positive.

This is shown in Fig. 1b-c.

They are positive for additional cell cycle markers too (**Point-by-point_Figure 3**).

Point-by-point_Figure 3: Cycling cells express *MKI67* and additional cell cycle markers (*CCNA2*, *CDK1*, *CCNB1*)⁶.

The interpretation of their data cannot be continued until they confidently achieve a situation in which they can without a doubt define cancer cells as such.

REPLY: Altogether, inferCNV analysis strengthens the unsupervised distinct clustering of tumor cells and normal keratinocytes, further supported by distinct enrichment for gene markers and published gene signatures. Of note, cycling cell cluster, which is composed of mixed tumor cells and keratinocytes was not considered for downstream analysis. Thus, we confidently identified and distinguished non-cycling tumor cells from non-cycling normal keratinocytes for further analysis.

They should demonstrate the CNV plots for all, and not only one case. My recommendation is also that they use only T cells as a reference for CNV inference in the test cells (e.g. cancer cells).

REPLY: As previously mentioned, the previous version of inferred CNV analysis integrated all tumor cells from the 5 BCC samples (and not only one case). We agree however with Reviewer #3 that it is

not ideal for inferCNV interpretation. As suggested, we now analyzed the non-cycling epithelial (*KRT14^{pos}*) cells from the 5 BCC samples individually, taking T cells as a reference (see Reviewer #3's first comment).

This is now shown in Fig. 1f and Supplementary Fig. 2a-d.

Second, perhaps the authors misinterpreted my initial suggestion "Third, virtually all analyses presented with the single-cell data have to be referenced against 1) patient of origin, 2) quality of cells (e.g. UMI or genes/cell count) to exclude trivial explanations for the variability they see among, e.g. low/intermediate/high-scoring genes for their infiltration signature." They address this by stating that they performed upfront quality control by filtering based on mitochondrial reads and gene complexity, and that those samples integrate (using stringent CCA). That is not the point. Even though they filter upfront, which is required, there remains high variability in quality within and across each patient (e.g. gene complexity may range from 500 to 6000 genes detected per cell as shown by the authors). Thus, upfront filtering will not be sufficient to control for variability in quality in specific analyses, where clustering of non-integrated data may be driven by within and across patient cell complexity. A simple way to address this is to show that clustering in multiple analysis is not associated with complexity (e.g. by projecting the range of UMIs/gene counts).

REPLY: We apologize for the initial misinterpretation. We understand Reviewer #3's concern that clustering may reflect within and across patient cell complexity, rather than biological processes in specific analyses. As suggested, we thus addressed the question by assessing the patient representation and gene expression complexity across both the tumor cells and fibroblasts subclustering.

Patient representation in tumor cells and fibroblasts once reclustered are now shown in Supplementary Fig. 6a and 10a. Except for tumor cell clusters 12 and 13 and fibroblast cluster 3 (in which 1 BCC sample was not represented), all BCC samples were represented across the various tumor cells and fibroblast clusters.

To further assess the potential impact of patient integration on clustering (and biological conclusions), we analyzed individual non-integrated BCC samples for the enrichment in $T^{NOD-TINF}$ and $S^{NOD-SINF}$ signatures in tumor cell and fibroblast subclusters respectively, that we present in the Point-by-point_Figures 4 and 5. By doing so, we observed similar ranges of $T^{NOD-TINF}$ and $S^{NOD-SINF}$ signatures enrichment in individual non-integrated samples (compared to integrated samples analysis), as well as a conserved negative correlation for the enrichment in $T^{NOD/TINF}$ and $S^{NOD/SINF}$ signatures respectively. This clearly argues against the clustering and the observed $T^{NOD-TINF}$ or $S^{NOD-SINF}$ gradual enrichment being dependent on patient integration.

Point-by-point_Figure 4: scRNA-seq analysis on tumor cell subpopulations of individual non-integrated BCC samples.

Point-by-point_Figure 5: scRNA-seq analysis performed on fibroblast subpopulations of individual non-integrated BCC samples.

Gene expression complexity was assessed by projecting the ratio $\log_{10}(\text{genes})/\log_{10}(\text{UMI counts})$ (https://hbctraining.github.io/scRNA-seq/lessons/04_SC_quality_control.html) in tumor cells and fibroblasts once reclustered. Importantly, we observed both satisfactory (ratio>0.8) and homogenous complexity across the various tumor cell and fibroblast subclusters.

This is now shown in Supplementary Fig. 6b and 10b.

To further assess the potential impact of complexity on clustering (and biological conclusions), we restricted our analysis to cells with virtually equal complexity (complexity ratio between 0.875-0.880 for tumor cells and between 0.865-0.885 for fibroblasts) (**Point-by-point_Figure 6**). By doing so, we observed a conserved negative correlation for the enrichment in $T^{\text{NOD}}/T^{\text{INF}}$ and $S^{\text{NOD}}/S^{\text{INF}}$ signatures in tumor cell and fibroblast clusters respectively. This clearly argues against the observed $T^{\text{NOD}}-T^{\text{INF}}$ or $S^{\text{NOD}}-S^{\text{INF}}$ gradual enrichment being dependent on gene expression complexity.

Point-by-point Figure 6: scRNA-seq analysis restricted to cells with highly similar complexity.
a) Tumor cells. b) Fibroblasts.

Of note, spatial T^{NOD} and T^{INF} as well as S^{NOD} and S^{INF} signatures were designed by opposition of distinct spatial compartments (spatially-resolved sequencing). The negative correlation observed for T^{NOD} and T^{INF} as well as S^{NOD} and S^{INF} signatures enrichment in tumor cells and fibroblasts clusters respectively clearly argues in favor of the biological relevance of the observed scRNA-seq clusters.

This is now shown in Supplementary Fig. 7a and 11a.

We also used a sequencing-independent method to further support the biological relevance of the scRNA-seq clusters. To do so, we selected identified marker genes for Low infiltrative morphology, High infiltrative morphology and collective migration and showed by RNA FISH (spatial mapping) the consistency of their spatial expression with the unsupervised clustering profile.

This shown in Supplementary Fig. 8a-c, Fig. 6d-e, Supplementary Fig. 12d, Supplementary Fig. 13a-f , Fig. 7b-c and Supplementary Fig. 15b-c.

Altogether, we now assembled various sequencing-dependent and sequencing-independent approaches that strongly argue in favor of a biological rather than technical clustering in our scRNA-seq analysis.

On behalf of all authors,

François Kuonen

References

1. Villani, R. *et al.* Subtype-Specific Analyses Reveal Infiltrative Basal Cell Carcinomas Are Highly Interactive with their Environment. *J. Invest. Dermatol.* (2021) doi:10.1016/j.jid.2021.02.760.
2. Kuonen, F. *et al.* TGF β , Fibronectin and Integrin α 5 β 1 Promote Invasion in Basal Cell Carcinoma. *J. Invest. Dermatol.* **138**, 2432–2442 (2018).
3. Marsh, D. *et al.* α v β 6 Integrin Promotes the Invasion of Morphoeic Basal Cell Carcinoma through Stromal Modulation. *Cancer Res.* **68**, 3295–3303 (2008).
4. Werner, S. & Alzheimer, C. Roles of activin in tissue repair, fibrosis, and inflammatory disease. *Cytokine Growth Factor Rev.* **17**, 157–171 (2006).
5. Hübner, G., Hu, Q., Smola, H. & Werner, S. Strong Induction of Activin Expression after Injury Suggests an Important Role of Activin in Wound Repair. *Dev. Biol.* **173**, 490–498 (1996).
6. Kashima, Y. *et al.* Combinatory use of distinct single-cell RNA-seq analytical platforms reveals the heterogeneous transcriptome response. *Sci. Rep.* **8**, 3482 (2018).

Reviewers' Comments:

Reviewer #1:

Remarks to the Author:

The authors have addressed all my comments

Reviewer #3:

Remarks to the Author:

The authors have diligently addressed my concerns.

They have already processed deposited data in GEO.

My only suggestion for the final version of this manuscript is to add a small section on "limitations" of the study, discussing such limitations, including small sample size. I think this is important to avoid future disagreements that may emerge with analyses of larger sample sizes with similar methods.

Manuscript Number: NCOMMS-21-33101B

“Integrated multi-omics reveals cellular and molecular interactions governing the invasive niche of basal cell carcinoma”

We thank the Reviewers for their contributive remarks.

Point-by-Point Rebuttal of Reviewer’s comments:

REVIEWERS' COMMENTS

Reviewer #1 (Remarks to the Author):

The authors have addressed all my comments

Reviewer #3 (Remarks to the Author):

The authors have diligently addressed my concerns.

They have already processed deposited data in GEO.

My only suggestion for the final version of this manuscript is to add a small section on "limitations" of the study, discussing such limitations, including small sample size. I think this is important to avoid future disagreements that may emerge with analyses of larger sample sizes with similar methods.

REPLY: As suggested, we mentioned the small sample size as a limitation of the study.

This is now mentioned in the Discussion section (page 20, lines 449-450).

François Kuonen